# High-resolution imaging of skin deformation shows that afferents from human fingertips signal slip onset

**Benoit P Delhaye[1,2]\*, Ewa Jarocka[3], Allan Barrea[1,2], Jean-Louis Thonnard[1,2], Benoni Edin[3], Philippe Lefèvre[1,2]**

[1]Institute of Information and Communication Technologies, Electronics and Applied Mathematics, Université catholique de Louvain, Louvain-la-Neuve, Belgium; [2]Institute of Neuroscience, Université catholique de Louvain, Brussels, Belgium; [3]Department of Integrative Medical Biology, Umeå University, Umeå, Sweden

**Abstract** Human tactile afferents provide essential feedback for grasp stability during dexterous object manipulation. Interacting forces between an object and the fingers induce slip events that are thought to provide information about grasp stability. To gain insight into this phenomenon, we made a transparent surface slip against a fixed fingerpad while monitoring skin deformation at the contact. Using microneurography, we simultaneously recorded the activity of single tactile afferents innervating the fingertips. This unique combination allowed us to describe how afferents respond to slip events and to relate their responses to surface deformations taking place inside their receptive fields. We found that all afferents were sensitive to slip events, but fast-adapting type I (FA-I) afferents in particular faithfully encoded compressive strain rates resulting from those slips. Given the high density of FA-I afferents in fingerpads, they are well suited to detect incipient slips and to provide essential information for the control of grip force during manipulation.

## Introduction

The most fundamental requirement for dexterous manipulation is the ability to handle objects without slippage and dropping of the object. To ensure that an object is held safely in the hand, one must apply a sufficient amount of force to the object's surface to counteract the forces tending to make it slip, for example, the object's weight and inertia (*Johansson and Flanagan, 2009*). Exerting an excessive grip is inefficient and can result in crushing the object. Inversely, a minimal amount of force is required to avoid slip and is dictated by friction: a stronger grip is required for slippery surfaces and a looser grip is sufficient for sticky surfaces. A good strategy is then to adjust the grip to friction with an amount of force slightly above the minimum. Previous work has suggested that humans can quickly and accurately adapt to changes in friction (*Westling and Johansson, 1984*; *Cadoret and Smith, 1996*). Importantly, tactile feedback is necessary for grip adjustments to take place as disruption of this feedback abolishes the normal, fine-tuned grasp control and results in frequent object dropping despite excessive compensatory grip forces (*Westling and Johansson, 1987*; *Augurelle et al., 2003*; *Witney et al., 2004*). However, how information about friction is encoded by tactile afferents is largely unknown.

Friction information might be partly available at the initial contact. Anecdotal evidence suggests that when contacting surfaces of different frictions the strength of the initial burst of activity of the afferents varies as the surface is changed (*Johansson and Westling, 1987*). In these experiments, however, different frictional conditions were associated with different surface textures and thus do not necessarily imply that the afferent responses specifically represented friction between the fingerpads and contact surface. Friction might not be encoded per se, but instead, tactile afferents might

**\*For correspondence:** delhayeben@gmail.com

**Competing interests:** The authors declare that no competing interests exist.

**Reviewing editor:** Cornelius Schwarz,

**eLife digest** Each fingertip hosts thousands of nerve fibers that allow us to handle objects with great dexterity. These fibers relay the amount of friction between the skin and the item, and the brain uses this sensory feedback to adjust the grip as necessary. Yet, exactly how tactile nerve fibers encode information about friction remains largely unknown.

Previous research has suggested that friction might not be recorded *per se* in nerve signals to the brain. Instead, fibers in the finger pad might be responding to localized 'partial slips' that indicate an impending loss of grip. Indeed, when lifting an object, fingertips are loaded with a tangential force that puts strain on the skin, resulting in subtle local deformations. Nerve fibers might be able to detect these skin changes, prompting the brain to adjust an insecure grip before entirely losing grasp of an object.

However, technical challenges have made studying the way tactile nerve fibers respond to slippage and skin strain difficult. For the first time, Delhaye et al. have now investigated how these fibers respond to and encode information about the strain placed on fingertips as they are loaded tangentially. A custom-made imaging apparatus was paired with standard electrodes to record the activity of four different kinds of tactile nerve fibers in participants who had a fingertip placed against a plate of glass. The imaging focused on revealing changes in skin surface as tangential force was applied; the electrodes measured impulses from individual nerve fibers from the fingertip. While all the fibers responded during partial slips, fast-adapting type 1 nerves generated strong responses that signal a local loss of grip. Recordings showed that these fibers consistently encoded changes in the skin strain patterns, and were more sensitive to skin compressions related to slippage than to stretch.

These results show how tactile nerve fibers encode the subtle skin compressions created when fingers handle objects. The methods developed by Delhaye et al. could further be used to explore the response properties of tactile nerve fibers, sensory feedback and grip.

respond to short and localized slip events that imply impending slip. There is indeed evidence that small, short-lasting slips occur during manipulation (*Johansson and Westling, 1984*). Those slips trigger strong afferent responses and elicit grip force adjustments (*Johansson and Westling, 1987*). In addition to the context of object manipulation, it has also been suggested that tactile slip detection plays an important role in a range of tactile tasks involving movements of surfaces relative to the skin (*Gueorguiev et al., 2016*; *Schwarz, 2016*).

Tactile slip is not instantaneous but develops progressively (*Levesque and Hayward, 2003*; *Tada et al., 2006*; *André et al., 2011*; *Terekhov and Hayward, 2011*; *Delhaye et al., 2014*; *Barrea et al., 2018*). As the surface-tangential force, that is, traction, of the fingerpad increases, 'local' slips begin at the periphery of the contact and progress toward the center until the last central 'stuck' point finally slips, that is, the instant of a 'full slip' or a global slip. We refer to the period between the beginning of the tangential loading and the instant of a full slip as the *partial slip* phase. Such a phenomenon gives rise to substantial local strain patterns in the slipping regions, near the boundary between the stuck and the slipping points (*Delhaye et al., 2016*). We hypothesized that information about these local deformations is carried by tactile afferents that inform the central nervous system about the contact state.

Single-unit recordings of primary tactile afferents, both in humans and monkeys, have shown that type I afferents respond strongly to local skin deformation (*Johansson et al., 1982*; *Sripati et al., 2006*; *Saal et al., 2017*). Those responses contain information about local geometric features such as edge orientation (*Pruszynski and Johansson, 2014*; *Suresh et al., 2016*; *Delhaye et al., 2019*). The most common stimuli used to evoke deformation of the skin are indentations and scanning with embossed geometric patterns or textures. Applying such stimuli makes it possible to relate the strength or the timing of the response to the topography or the statistics of the stimulus itself, but does not provide a mechanistic understanding of the nature of the response with respect to the local skin deformation at the mechanoreceptors themselves. Moreover, it is mostly unknown how tactile afferents respond to surface strains, that is, strains acting tangentially to the surface (as opposed to features indented perpendicularly to the skin surface). Afferent recordings in the hairy skin of the

human hand have shown that afferents of all types strongly respond to local skin stretch and that the fast-adapting type I (FA-I) afferents also strongly respond when the stretch is released (*Edin, 1992*; *Edin, 2004*). Slowly-adapting type II (SA-II) afferents are also known to be sensitive to skin stretch, but relating their response to the exact local stretch pattern is complex given the large size of their receptive fields. How glabrous skin afferents, that is, those engaged in the contact with objects during manipulation, respond to local skin strain has, to our knowledge, never been studied. This is mainly due to the difficulties of applying well-controlled mechanical stimuli and measuring the strain at the same time.

To address this, we took advantage of a recently developed imaging system that can measure fingertip skin strain through a transparent material during tangential loading of the fingertip until slip occurs (*Delhaye et al., 2014*; *Delhaye et al., 2016*). While recording *local* strains with this system, we simultaneously recorded the activity of human tactile afferents innervating the fingertip (*Figure 1A*). This way we were able to relate local strains to responses of afferents with receptive fields inside the fingerpad contact area. We found that all tactile afferents in the fingertip responded to slip events, but that FA-I afferents in particular faithfully signaled local skin compressions related to the progression of slips. We suggest that FA-I afferents are primarily responsible for detecting changes in surface strains and that their discharges are a primary source of information for the central nervous system to, for instance, quickly adjust fingertip forces to different levels of friction.

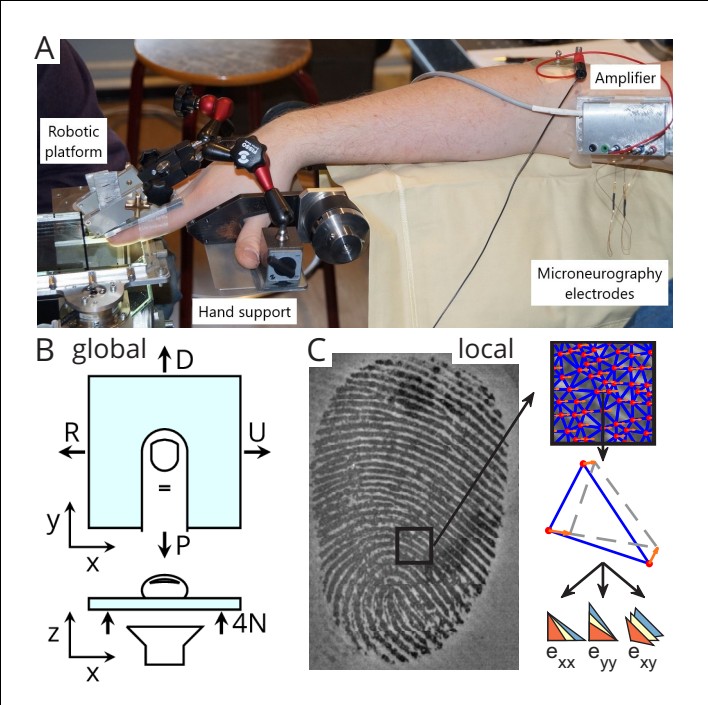

**Figure 1.** Experimental setup. (**A**) A robotic platform (left) was used to move a transparent plate of glass in contact with the fingerpad while the responses of single tactile afferents were recorded from the median nerve using microneurography (right). (**B**) The plate exerted a servo-controlled normal force of 4 N and was moved tangentially in one of four directions (U: ulnar; D: distal; R: radial; P: proximal). At the same time, a camera was used to image the contact area. All fingerprint images and strain heatmaps are shown using the same view, with the ulnar on the right. (**C**) From the fingerpad images, features (red dots) were sampled and tracked from frame to frame (the orange arrows show the features motion to the next frame). Features were then triangulated and the triangle strains were computed, leading to two axial strain components ($e_{xx}$ and $e_{yy}$) and a shear strain component ($e_{xy}$). Lower-right pictograms show how an initial triangle (in yellow) is deformed when experiencing positive (tensile, in blue) or negative (compressive, in red) strain.

## Results

Slips were applied to the fingerpad using a robotic platform holding a transparent plate of glass that was either plain, yielding high friction, or covered with a hydrophobic coating yielding a lower friction. The plate first made contact with the fingerpad ('contact' in *Figure 2A, C*) and then moved tangentially at constant velocity until full slip in one of four different directions: ulnar, distal, radial, and proximal, and then moved back in the opposite direction until full slip occurred again (forward and backward, respectively, *Figure 1B*, *Figure 2A, C*). The normal component of the force was servo-controlled to be kept at 4 N, and the tangential component was developed as a consequence of the tangential movement of the plate. At the same time, we imaged the fingerpad contact and tracked numerous features on fingertip ridges as the slip progressed (*Figure 1C*). Finally, the plate was moved down ('release' in *Figure 2A, C*). We were able to precisely monitor skin strains from frame to frame (i.e., change in strain or strain rates) in the contact area during the transition from a fully stuck contact to a fully slipping contact (see also *Delhaye et al., 2016*). The skin strains were measured in the contact plane and expressed in terms of three independent components: two axial components aligned to the plate movements ($e_{xx}$ and $e_{yy}$) and one shear component ($e_{xy}$, *Figure 1C*). Importantly noted, the presence of local strains also indicates that the skin is locally slipping. That is, the limit between the stuck and slipping region is just preceding the front of the strain waves.

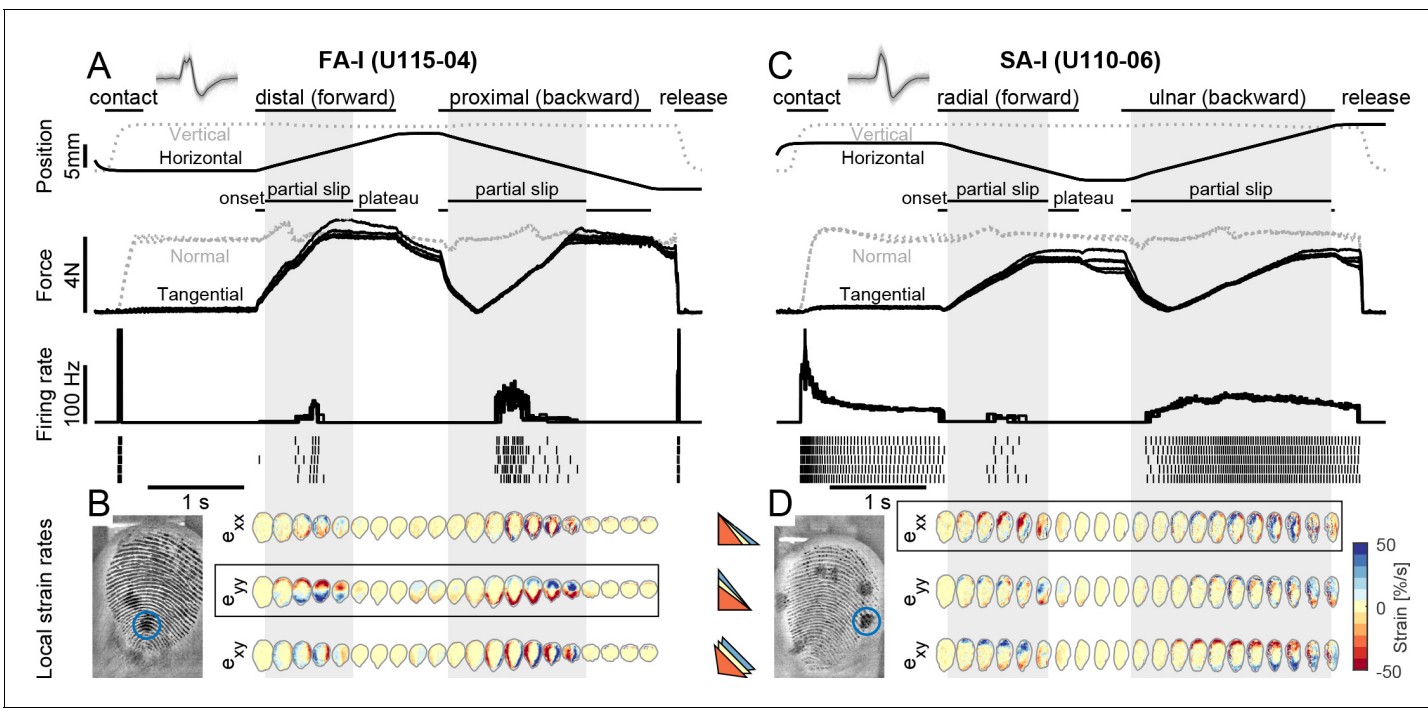

**Figure 2.** Experimental procedures and typical traces. (**A**) Evolution of the global variables, the plate position (vertical in gray and horizontal in black) and the contact force (normal in gray and tangential in black), together with the afferent instantaneous firing rate (and the spikes) as a function of time for a typical fast-adapting type I (FA-I) unit in the high friction condition. The plate was pressed against the fingerpad ('contact'), moved tangentially forward until the occurrence of a full slip, then moved backward, and finally retracted ('release'). The partial slip phase is highlighted by the gray vertical boxes. Five repetitions are overlaid. The tangential movement was split into three phases: onset (lasting 100 ms), partial slip, and plateau. (**B**) Heatmaps of the evolution of the local surface strain rates in the contact area as a function of time during the tangential loading, for one of the five repetitions. The three strain components, axial along x and y, and shearing, are shown (colored triangles depict the deformation axes). Compression (negative strain) is in red. The location of the unit receptive field center is shown with a blue circle on the raw fingerpad image on the bottom left. (**C, D**) Same as in (**A, B**) for a typical slowly-adapting type I (SA-I) unit. For both units, insets show recorded, superimposed action potentials and their average shape represented by a dark line.

The online version of this article includes the following figure supplement(s) for figure 2:

**Figure supplement 1.** Instantaneous firing rate as a function of time for all units recorded and all conditions.

**Figure supplement 2.** Correlation of firing rates with contact forces.

We focused the analyses on tangential loading movements, from the moment when the plate started to move tangentially and until it completed a forward or a backward movement. Each tangential loading movement was split into three sequential phases defined as follows: (1) the *movement onset phase* arbitrarily defined as the initial 100 ms; (2) the *partial slip phase* that lasted until the tangential force reached a plateau and the finger fully slipped; and (3) the *plateau phase* during which the finger was fully slipping against the glass surface and that lasted until the end of the tangential movement (*Figure 2A, C*). Strain changes were observed in the contact area during the partial slip phase in the form of two waves of opposite signs (*Figure 2B, D*). Remember that the strain wavefront is where the slip starts to occur. Those waves started moving at the onset of movement from opposite sides and from the periphery of the contact toward the center and disappeared at the point of full slip (*Delhaye et al., 2016*). Once the full slip was reached, the changes in strain faded away. The components of the strain changes corresponding to the direction of the plate movement were the largest in amplitude (*Figure 2B, D*, circled by a black box). Different movement directions elicited different patterns (i.e., compression, stretch, or shear) at all points in the contact area. For instance, the receptive field of a given afferent could be stretched along the proximal-distal axis for movements in the distal direction but compressed along the same axis for movements in the opposite (i.e., proximal) direction (*Figure 2B*, $e_{yy}$).

## Tactile afferents strongly respond during partial slip

We used microneurography (*Vallbo and Hagbarth, 1968*) to record the activity of single units whose receptive fields were located at the fingertip (*Figure 1A*). We focused on FA-I and SA-I afferents, which respond to local deformation events and have small, well-defined receptive fields (*Johansson and Vallbo, 1983*). We also recorded from a few type II afferents (FA-II and SA-II). Sufficient recordings for data analysis (three out of five repetitions of each plate move direction) were obtained from 22 afferents (13 FA-I, 6 SA-I, 2 SA-II, and 1 FA-II). The locations of the receptive fields of all afferents are depicted in *Figure 2—figure supplement 1*. As expected, the afferents responded vigorously to contact (*Figure 2*, 'contact'), but also responded in a variety of ways to the tangential loading (*Figure 2*, 'forward' and 'backward'). First, we looked at the overall discharge pattern of the afferents. FA-I units showed a phasic response, with a burst of activity at the instant of contact, and another one during the tangential loading (e.g., *Figure 2A*). However, a majority of FA-I afferents responded mostly only during the partial slip phase, showing no or almost no response during the start ('onset' phase) and the end ('plateau' phase) of the tangential movement (for U115-04 in *Figure 2A*, there was one spike at the onset phase during one repeat). SA-I units instead presented a rather tonic response beginning at the initial contact that changed during the tangential loading phase by increasing or decreasing their firing rates (e.g., *Figure 2C*). The discharge patterns of all recorded afferents for all directions and frictions are reported in *Figure 2—figure supplement 1*. *Video 1* and *Video 2* show image recordings of the fingerpad during one trial, together with spike sound associated with the afferent responses, for the two example units shown in *Figure 2*.

Note that the tangential movement of the plate led to slight fluctuations in the normal force that could not be suppressed by the force controller (see Materials and methods). Those fluctuations did not evoke strong afferent responses. Indeed, the discharge rates were neither correlated to the normal force nor to its derivative (*Figure 2—figure supplement 2*). In fact, we considered a causal relationship untenable observing in *Figure 2A* that the afferent discharge seemed to follow the normal force

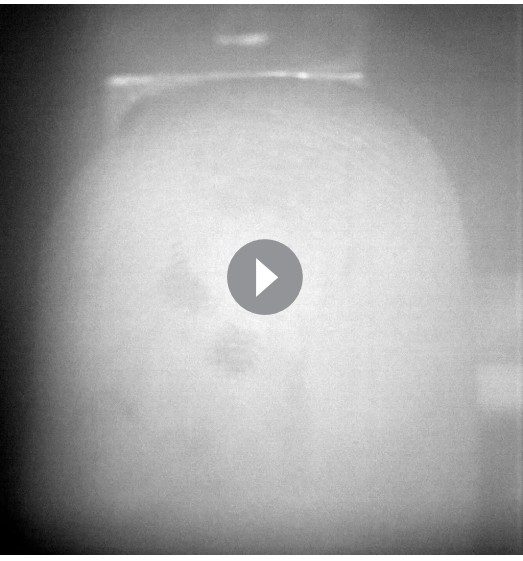

**Video 1.** Image recordings of the fingerpad during one trial (distal, high friction), together with spike sound associated with the afferent responses (unit 115-04).
https://elifesciences.org/articles/64679#video1

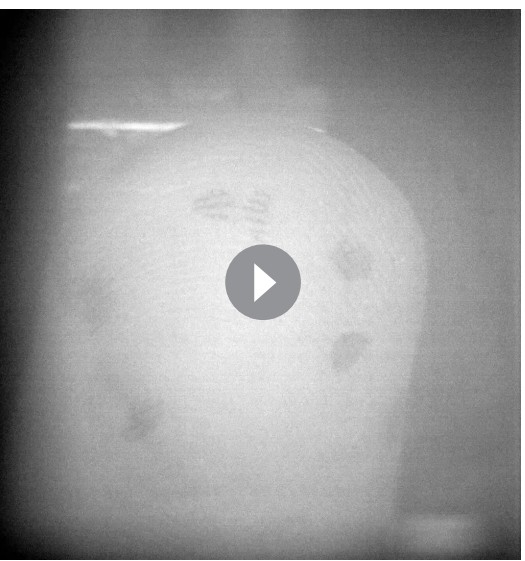

**Video 2.** Image recordings of the fingerpad during one trial (radial, high friction), together with spike sound associated with the afferent responses (unit 110-06). https://elifesciences.org/articles/64679#video2

fluctuation by ~100 ms in the distal direction but preceded it by ~200 ms in the proximal direction.

First, we describe how the tactile afferent responded with respect to 'global' stimulus parameters such as the movement phase or direction. Afferents were more active during the partial slip phase than during the two other tangential loading phases (i.e., onset and plateau), suggesting that this period is key to the afferent responses (*Figure 3A*). Indeed, for all four afferent types, the fraction of trials during which the afferent responded with at least one spike was higher for the partial slip phase than for the two others (one-way ANOVA with repeated measures, F(2,24) = 47.05, p<0.001 for FA-I units; F(2,10) = 13.97, p=0.001 for SA-I units; all p<0.05 for paired comparisons; for FA-II and SA-II afferents, we did not have enough data for such analysis). FA-I units were silent during the plateau, except when a stick-and-slip phenomenon was observed, in which case the firing phase was locked to the stick-and-slip events (e.g., see U109-04 during plateau phase in *Figure 2—figure supplement 1*).

For each afferent, we defined its preferred *global direction* (i.e., ulnar, distal, radial, or proximal), namely, the plate movement direction for which the highest mean firing rate was observed during partial slip. *Figure 3B* shows the mean firing rate in the preferred *global* direction (labeled 'North') and in the remaining other directions with respect to the preferred one. We found that different movement directions elicited different responses, confirming previous observations (*Birznieks et al., 2001*). Some FA-I units tended to have a preferred-opposite pattern (5 out of 13), that is, the preferred ('N' for *North* in *Figure 3B*) and opposite directions ('S' for *South* in 3B) tended to elicit a stronger response than the two other directions. This was not observed for the other afferent types, which tended to discharge less in the direction opposite to the preferred direction. The distribution of the preferred *global* direction covered the four directions tested; however, we observed overall a slight preference for the proximal-distal axis (*Figure 3C*). Firing rates during partial slips were consistent across forward and backward (*Figure 3D*; correlation r = 0.97, n = 176, p<0.001), suggesting similar directional preference when the skin was already under tension (backward movement). While the firing rates were strongly correlated across friction levels (*Figure 3E*; correlation r = 0.95, n = 176, p<0.001), the firing rates across the whole population tended to be slightly lower for low friction (paired t-test, t(171) = 7.57, p<0.001) and the ratio of the mean firing rate during low and high friction condition was 0.82 ± 0.55 (mean ± std).

## FA-I afferents respond to local skin strain rates

Next, we sought to relate the afferent discharge to the strain pattern, that is, the local events taking place inside the afferent receptive field. We observed that the discharge of FA-I units was strongly coupled to the compressive strain changes taking place in their receptive fields during the partial slip phase (*Figure 4*). This was particularly clear for U104-02, whose receptive field (depicted with a gray circle in the fingerpad heatmaps of *Figure 4*) was located on the proximal side of the contact area. Indeed, during a proximal movement (*Figure 4A*, left, and B, right), the plate movement elicited a compressive wave of strain changes (in red) along the y-axis (proximal-distal axis) moving from the proximal side of the contact area toward the center and crossing the afferent's receptive field. This crossing elicited a strong discharge burst. Moreover, the low friction surface exhibited full slip at a lower level of tangential force, and therefore also earlier than the high friction surface, with respect to the movement onset. As a consequence, the compressive strain wave moved across the afferent receptive field earlier in this case and, strikingly, the discharge burst of the afferent also

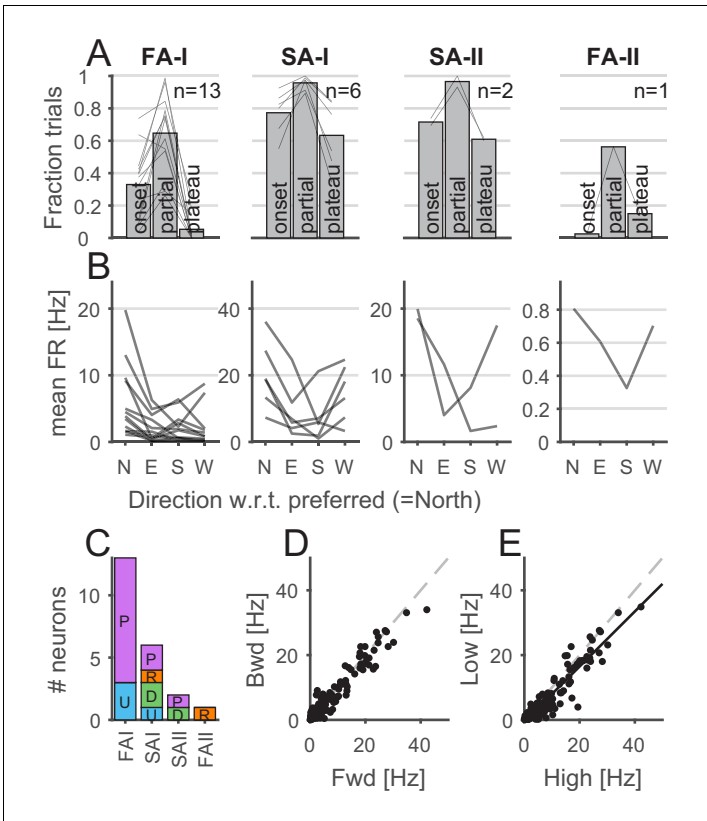

**Figure 3.** Properties of afferents' responses during tangential loading. (**A**) The afferents were mostly active during the partial slip phase. The fraction of trials during which the afferents were active for each phase of the tangential loading (onset, partial, and plateau) for the forward and backward movements. Bars show the average across units, and the lines show individual afferents (n = 13, 6, 2, and 1 for fast-adapting type I [FA-I], slowly-adapting type I [SA-I], slowly-adapting type II [SA-II], and fast-adapting type II [FA-II], respectively). (**B**) Afferents discharge more in a given direction. Mean firing rates elicited by partial slip as a function of angle with respect to the preferred direction ('North') for each unit. Each line shows a different afferent, and the mean firing rate was averaged across repetitions, movements ('fwd' or 'bwd'), and frictions. (**C**) Distribution of the afferent preferred *global* direction for each afferent type. (**D**) Mean firing rate for backward versus forward movements. One data point is shown for each afferent (n = 22) and each condition (8 = 4 direction × 2 friction) and averaged across repetition. The dashed gray line is the slope = 1. (**E**) Mean firing rate for low versus high friction. One data point is shown for each afferent (n = 22) and each condition (8 = 4 directions × 2 forward-backward) and averaged across repetition. The black line is the least square regression, and the dashed gray line is the slope = 1.

took place earlier, coinciding with the strain changes. This is even easier to observe for the backward movement (*Figure 4B*, right). In this case, due to the previous loading, the movement of the compressive wave came even earlier when the low friction condition was used and the discharge burst of the afferent coincided. Finally, we observed that the response evoked by a stretch wave, generated by a distal movement (*Figure 4A*, right, and B, left), was much weaker than its compressive counterpart. Still, the timing of the response was perfectly synchronized with the occurrence of the stretch in the receptive field. Note that a short burst was elicited at the onset of the movement in the distal direction in the backward case (*Figure 4A*, right). Such transient burst cannot be explained by our strain measurements and occurred in a small fraction of the trials and only in a few afferents (*Figure 3A*). Also note that part of the unit receptive field lost contact during the partial slip phase in the high friction case.

To test the hypothesis that the responses of the tactile afferents are caused by specific 'local' strain patterns taking place inside the contact area, we took two different approaches. In the first model-free approach, we looked at the strain pattern observed at the time of each spike across all stimulus directions and frictions and computed a 'spike-triggered average' (STA, see

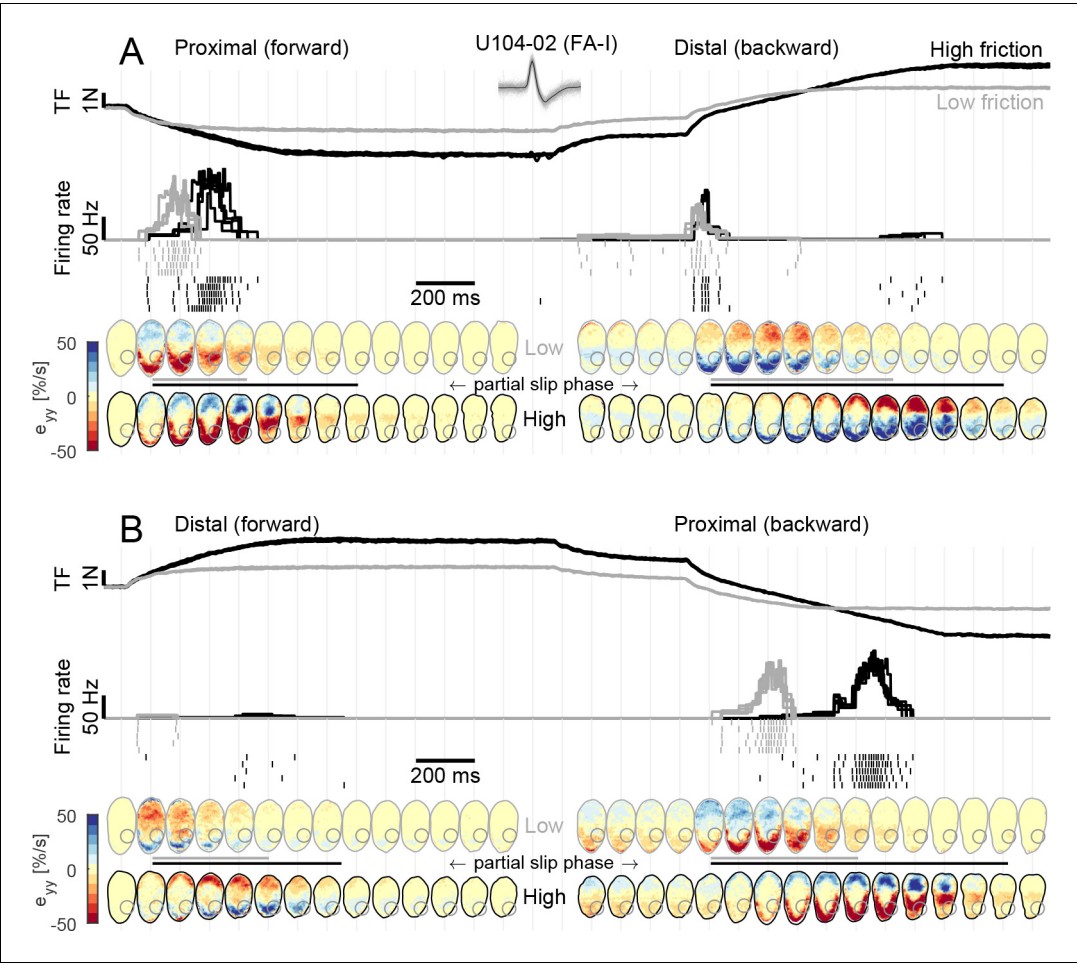

**Figure 4.** Fast-adapting type I (FA-I) responses during partial slip are related to local strain rates. (**A**) Evolution of tangential force, afferent instantaneous firing rate (together with the spikes), and strain rates as a function of time during tangential loading in the proximal direction (forward) followed by a distal movement (backward). Data are shown for two different frictions, high in black and low in gray, and are aligned on the onset of movement. Five repetitions are shown, except for the strains for which one trial is shown. The compressive strain (negative) is in red. The color of the contact area contour indicates the friction condition, and the gray circle shows the location of the afferent receptive field. The horizontal lines between the heatmaps depict the partial slip phase as shown in *Figure 2*. (**B**) Same as in (**A**) but for distal movement (forward) followed by a proximal movement (backward). Inset shows recorded, superimposed action potentials and their average shape represented by a dark line.

Materials and methods) for all recorded FA-I and SA-I units. If the afferents with a receptive field in the contact area responded to local strains, we expected to observe a clear strain pattern associated with these units' discharges. In contrast, for afferents with a receptive field outside the contact area, we expected no clear strain pattern at all. First, we used the strain rate norm ($\|e\|$) as a variable to estimate the STA. We found that, indeed, the average strains causing spikes in all recorded FA-I afferents had a clear, more or less annular (ring-like) pattern (*Figure 5A*). Such an annular pattern is expected from the stimulus, which is a strain wave in the form of an annulus and does not reflect the shape of the afferent receptive field but rather the correlations present in the strain patterns (Materials and methods). Importantly, however, the pattern overlapped the afferent's receptive field and often peaked inside it. Furthermore, as expected, we did not observe such a clear pattern for the afferents having their receptive field outside the contact area (*Figure 5*, middle). Strikingly, clear patterns did not emerge for the SA-I afferents, suggesting that those afferents are less sensitive to the local surface strain changes (*Figure 5A*, bottom). Note that we repeated the same analysis using the total (cumulative) strain instead of the strain changes, and that again, we did not observe any clear pattern. The peak values of the STA are shown in *Figure 5B* in orange and show strong values

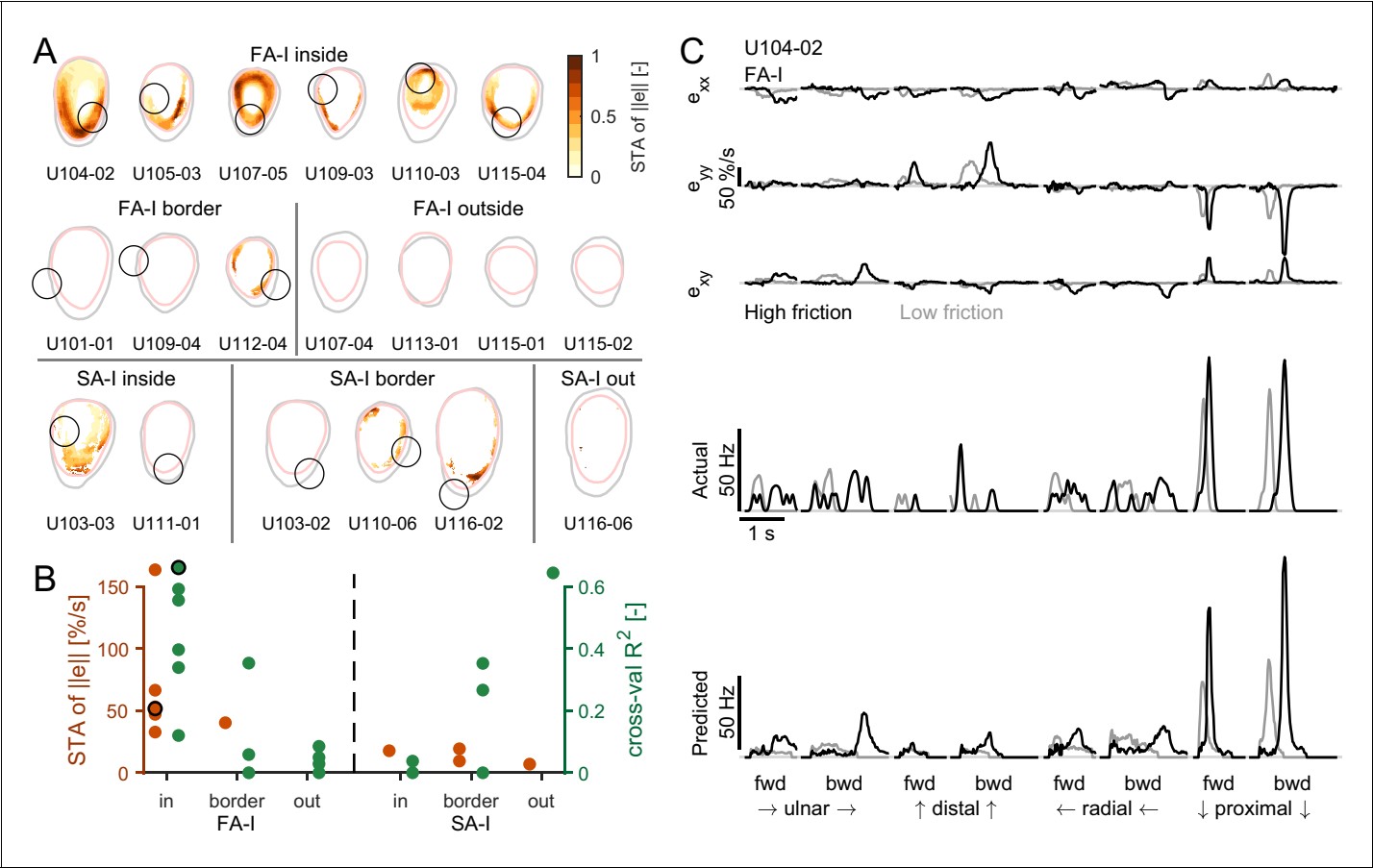

**Figure 5.** Afferent responses to surface skin strains. (**A**) Heatmaps of the spike-triggered average (STA, in orange) for the fast-adapting type I (FA-I) units inside the contact area (top row), the other FA-I units (middle row), and the slowly-adapting type I (SA-I) units (bottom row). The STA matrices were obtained from the norm of the strain ($\|e\|$) and normalized to the maximal value (reported in **B**). The gray contour shows the initial contact area, and the pink contour depicts the parts of the contact area that remained in contact for at least 50% of the partial slip phase. The black circle is the unit receptive field. (**B**) STA peak value (red) and linear regression model performance (green). For units in or on the border of the contact area, the maximum was taken inside the receptive field. For units outside the contact area, the maximum was taken anywhere inside the contact area. The STA of some units were undefined over the whole contact area (see Materials and methods); those units are not shown. The unit shown in (**C**) is highlighted with a black circle. (**C**) Time evolution of the strain rates inside the receptive field, the actual firing rate (represented as spike events convolved with a Gaussian kernel, see Materials and methods), and the predicted firing rate of an example unit during partial slip for each direction and each friction (black is high friction, gray is low friction).

The online version of this article includes the following figure supplement(s) for figure 5:

**Figure supplement 1.** The same heatmaps as in *Figure 5A* but showing the spike-triggered average (STA) for the two principal strain rates ($e_1$, compressive, and $e_2$, tensile).

**Figure supplement 2.** The same heatmaps as in *Figure 5A* but showing $R^2$ for friction, direction, and forward/backward cross-validation methods.

for FA-I units inside the contact and much weaker values for other afferents, confirming the previous observations. The same STA analysis was repeated using the two principal strain components, one compressive and one tensile, separately to build two STA maps (see Materials and method). The results obtained are consistent with the STA obtained with strain rate norm, that is, that a clear pattern emerges only for FA-I afferents and that the STA peaks in the afferent receptive field (*Figure 5—figure supplement 1*). Moreover, we found that the compressive STA peaks were generally larger and more often found in the afferent receptive field than their tensile counterpart, suggesting that FA-I afferents are more sensitive to compression. This finding will be further supported in the next section.

In the second, model-based approach, we aimed to predict the afferent discharge rate from the skin strain measured in the contact area. Inspection of the data led us to assume that, first, strains of

opposite signs might not contribute in the same manner to the discharge as skin stretch seemed to evoke weaker responses than skin compression (*Figure 4*). Second, multiple components might be needed to explain responses in all directions. Therefore, we first set out to test if the afferents' firing patterns could be reliably predicted for the skin strains using a model with six distinct predictors, that is, the three strain components each half-wave rectified, using both the positive (stretch) and negative part (compression). The simplest model possible, a multiple linear regression including an intercept, was used. Our method was cross-validated, such that the regression models were fit on one friction condition and tested on the other (Materials and methods). In the subsequent validations, we used data from different directions and forward vs. backward movements and observed quantitatively similar results (see *Figure 5—figure supplement 2*). The results obtained using the model-based approach revealed similar trends as the first model-free method. The linear model could predict the discharge pattern of the FA-I afferents, but not the SA-I afferents. Heatmaps built from the cross-validated $R^2$ (*Figure 5—figure supplement 2*) were qualitatively similar to those built from the STA. An example unit is shown in *Figure 5C* for an afferent with a substantial $R^2$ (0.66). This unit maximally responded in the proximal direction, when a compressive wave was observed in its receptive field. The maximal values of the $R^2$ found within the receptive field (or anywhere for the afferents outside the contact area) are shown in *Figure 5B* in green. As with the model-free approach, only the firing rates of FA-I *inside* the contact area could be predicted from the strain. The high $R^2$ value for the SA-I afferent outside the contact area is because this particular afferent was either active at a constant firing rate or silent, generating two separate clouds of data points and thus driving up the $R^2$ (*Figure 2—figure supplement 1*).

In summary, we observed that FA-I afferents respond to local strain patterns generated during partial slips.

## Aspects of the skin strain rates encoded by the FA-I afferents

Having demonstrated that the recorded FA-I afferents respond to local strain patterns, we then sought to uncover what aspects of the strains were responsible for these responses. To that end, we aimed to predict FA-I afferent discharge rates with a subset of strain predictors and to compare their performance to the full model with six predictors. The analysis was performed only on the FA-I afferents for which we had optical measurements of the strains, that is, those having their receptive fields inside the contact area (n = 6, all shown in the top line of *Figure 5A*). Since all models are cross-validated, they can be compared irrespective of the number of predictors. First, we selected three single predictors that were invariant to the choice of a particular reference frame. The strain norm, informative about the intensity but neither the orientation nor the sign of the deformation, and the two principal strain components separately, the compressive ($e_1$) and the tensile ($e_2$), obtained from single-value decomposition of the strain tensor (Materials and methods), informative about the intensity and the sign of the deformation, irrespective of the orientation of the deformation. All those three single predictor models performed worse than the full model, as could be expected (*Figure 6A*, left). However, we found that the compressive principal component always outperformed the tensile one, suggesting that the afferents are more sensitive to compression than to stretch.

Next, we used each of the predictors of the full model separately. That is, the half-wave rectified positive and negative value of the three strain tensor components ($e_{xx}$, $e_{yy}$, and $e_{xy}$). Given that those components are dependent on the choice of a particular reference frame orientation, we repeated the fitting procedure for multiple rotations of the reference frame equally spaced from 0 to 90° (Materials and methods). The results are shown in *Figure 6B* for an exemplar afferent, with the shear component ignored. In this figure, the performance of the prediction ($R^2$) based on a single strain component (compressive in red and tensile in blue) is shown as a function of the reference frame rotation. This afferent seemed to have a preferred strain orientation close to 90° with respect to the radial-ulnar axis (i.e., along the proximal-distal axis), where the compressive component peaks. That is, the afferent seemed to encode preferentially 'local' compressive strain rate along a particular orientation. Indeed, as already described in *Figure 4*, this unit was responding strongly in the proximal condition, where a compressive strain wave along the proximal-distal orientation passed through its receptive field. Perpendicular to that orientation, the tensile component peaked as well but with a lower $R^2$. This is expected since compression in one orientation generates stretch in the other at the same time because the volume is mostly conserved. The same plot as in *Figure 6B* is provided for

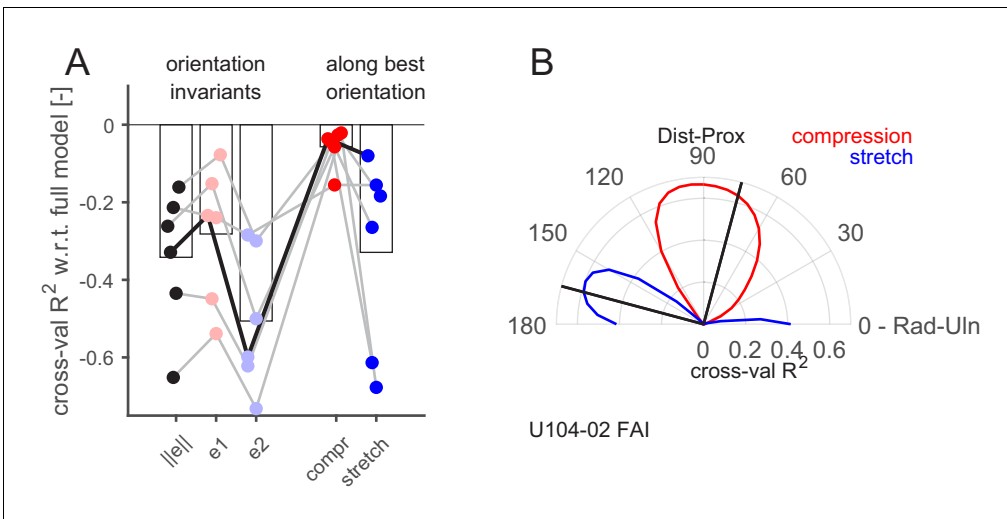

**Figure 6.** Aspect of the skin strain encoded by the fast-adapting type I (FA-I) afferents inside the contact area. (**A**) Firing rate prediction performance (cross-validated $R^2$) for different models with respect to the full model comprising six predictors (0 on the y-axis corresponds to the full model performance, lower is worse). The three first models ($\|e\|$, $e_1$, and $e_2$) are single predictors and rotation invariant. The two last models (compression and stretch) have one component, and the performance obtained with the best orientation is shown. The unit shown in (**B**) is highlighted with a black line. (**B**) Polar plot showing the performance of the prediction (cross-validated $R^2$) of linear models based on single strain components (red for compression and blue for stretch) as a function of the rotation angle of the reference frame (from 0 to 90°) with respect to the radial-ulnar axis (0°). Data from an example afferent U104-02. The black lines show the frame rotation that yields the best performance (maximizing the sum of the $R^2$ of the compressive and tensile models).

The online version of this article includes the following figure supplement(s) for figure 6:

**Figure supplement 1.** The same polar plots as *Figure 6B* but for all units and all cross-validation methods.

**Figure supplement 2.** Relation between preferred strain rate orientation and fingerprint ridges orientation.

all FA-I afferents and the three cross-validation methods in *Figure 6—figure supplement 1*. It is important to avoid the confusion between the units' preferred *direction* described in *Figure 3*, which relates to the robot movement direction and the maximal firing rate of the afferent, and the *strain orientation* preference described here, which is related to the orientation of the local deformation in the afferent receptive field.

From these analyses, we draw two important conclusions. First, compressive strain change is a more effective stimulus than tensile strain change (*Figure 6A*, right), confirming the observation in the principal component analyses, namely, that the FA-I responses are mostly driven by compression. Second, FA-I afferents did not respond to compression in any orientation but rather to compression along a certain preferred strain orientation. The argument for this is twofold: (1) the performance of a single compressive component in a particular orientation was always higher than the performance of the compressive principal component $e_1$ (*Figure 6A*) and (2) models with one single component along its preferred strain orientation performed as well as the full model comprising all the components, suggesting that this component is mainly responsible for driving the afferent response. Note that the FA-I afferents' preferred strain orientation seemed to coincide with the local fingerprint orientation, but more data is needed to confirm this trend (*Figure 2—figure supplement 2*, correlation r = 0.74, p=0.10, n = 6).

In sum, our analyses strongly suggest that the FA-I are sensitive to local skin strains, more so to changes in compressive strain than tensile strain and that they respond maximally along a preferred strain orientation.

## SA-I and SA-II responses are related to external forces

Finally, we asked how much the response of all types of afferents was related to the 'global' external 3D force vector. We computed the correlation between the afferent firing rates and each force

components during the tangential loading (all three phases). We also computed the same correlations with the force rates. We found low correlation values for both force and force rates for the fast-adapting afferents (FA-I and FA-II). However, the correlations with tangential (horizontal) forces were high for slowly-adapting afferents (SA-I and SA-II), especially for those not inside the contact area (*Figure 2—figure supplement 2*). Their firing rates were however not related to the small vertical (normal) force fluctuations.

## Discussion

This is, to our knowledge, the first study that investigates the relationship between the responses of tactile afferents to slippage and the detailed surface strain patterns in glabrous skin. Specifically, the results from our analyses imply that the local compression induced by partial slip generates strong responses from FA-I but not SA-I units. Given that the FA-I from the fingertip are mostly silent when the fingerpad is stuck to the object and even at the onset and offset of the plate movement, their vigorous responses to the changes in compressive strain provide a particularly salient signal that enables contact stability. It is reasonable to suspect that FA-I afferents play a crucial role in maintaining grasp stability since they constitute more than 50% of all afferents in the glabrous skin of a human fingertips (*Johansson and Vallbo, 1979*). Our results show that while partial slips give rise to strong responses, the friction per se is poorly encoded by FA-I. Finally, given that cortical somatosensory neurons are strong edge detectors (*DiCarlo et al., 1998*; *Delhaye et al., 2018*), the particularly salient signal from the FA-I afferents will likely be amplified downstream.

A previous study showed that information about grip safety, that is, the increment of tangential force needed to reach full slip, is present in the response of local tactile afferents at the level of a population of afferents, irrespective of the level of friction (*Khamis et al., 2014*). Given that all afferents respond to progressing slips, and that the timing of each FA-I afferent's burst is dependent on how far from the center of contact the afferent is, it is not surprising that a linear combination of multiple FA-I units can provide a good estimate of the grip safety. While this previous study demonstrated that the information is present in the response of the population, our approach provides a mechanistic explanation and underlines the saliency of the grip safety signal.

An essential component of this work was the use of two different frictions. By using the same protocol with two different frictions, we were able to disambiguate the relative contribution of the external force and the local strain measurements to the afferent response. Indeed, until an instant very close to full slip in the low friction condition, the low and high friction materials lead to very similar normal and tangential force profiles, whereas the surface strain patterns inside the contact area have different timing and their progressions differ very early on. Thanks to this contrast, uniquely provided by the difference in friction, the earlier afferent response in the case of the low friction cannot be attributed to the change in tangential force itself. The fact that the afferent responses, particularly the FA-I units, are related to local slip and are therefore poor predictors of friction per se suggests that humans could be exquisitely sensitive to detect slip while being poor at estimating friction. That is, estimating friction would require the occurrence of partial slips, and the absence of slips would make such estimates impossible.

The FA-I afferent responses were best predicted by compressive strain, rather than tensile strain. Moreover, the afferents were mostly sensitive to a particular strain orientation. This property might be explained by the shape of the afferent receptive fields, which is elongated (rather than circular) and contains numerous zones of high sensitivity ('hot spots', *Johansson, 1978*; *Phillips et al., 1990*; *Jarocka et al., 2021*). A particular arrangement of those hot spots might lead to increased sensitivity to a particular strain orientation. Indeed, it has been demonstrated that particular arrangements enable each afferent to differentiate the orientation of edges moving across the receptive fields from the strength or the fine timing of their response (*Pruszynski and Johansson, 2014*; *Suresh et al., 2016*). In fact, the moving strain wave related to partial slip observed in this study can be seen as an edge moving across the contact area.

### Limitations

Our methods generate high-resolution representations of the *distribution* of surface strain in the contact area of the fingertip (i.e., 2D, x and y axes) but do not allow measurements of the *distribution* of the deformations perpendicular to the fingertip surface inside the contact area (i.e., along

the z-axis), nor outside the contact area. Given that the SA-I afferents did not respond to the surface deformation measured in this study, they could be sensitive to two other non-measured aspects of the local deformations. First, since SA-I afferents are known to be exquisitely sensitive to contact and contact pressure (*Goodwin and Wheat, 1999*; *Wheat et al., 2010*; *Khamis et al., 2015*), they may be more sensitive to the strains perpendicular to the surface, rather than the tangential (i.e., planar) strains. Indeed, the tangential loading can cause a progressive redistribution of the pressure inside the contact area (even if the global normal force is maintained constant) that will consequently generate 'local' vertical strains (not measured, which would be denoted '$e_{zz}$'). The SA-I shows slow firing rate variations that might be well attributed to a relatively slow movement of the center of pressure during tangential loading that results in local pressure change. Another possibility is that the SA-I afferents are sensitive to local shear (which would be denoted '$e_{xz}$' and '$e_{yz}$'). The fact that their firing rates correlate with the tangential forces supports this idea, and more measurements are needed to disentangle the two possibilities. Importantly, these two other aspects of skin deformations are likely much less informative about the partial slip state of the finger; therefore, SA-I afferents are less likely to contribute to the detection of the slip events. When considering the subdermal location of SA-II and FA-II afferent terminals, it is not surprising to observe a poor coupling between their responses and the measured surface strains.

The tangential speed chosen for practical reasons in our experiments is certainly lower than that during normal manipulation actions, during which much higher strain rates are probably generated. In fact, it is unknown what the typical amplitudes of surface skin strain during manipulation are. Moreover, during active manipulation, humans typically also vary the amount of grip (normal) force according to the tangential force, while our experiment kept the normal force constant. The development of a manipulandum equipped with embedded imaging of skin deformation in the contact area will make it possible to quantify deformation in an ideal environment with both normal and tangential force fluctuations (*Barrea et al., 2017*). Nevertheless, it is reasonable to expect that FA-I afferents respond even more vigorously during actual manipulation. It remains also to be demonstrated that in a typical manipulation task the timing of this signal enables online control of grip, that is, that there is enough time to react. In this study, the partial slip phase lasts several hundreds of milliseconds, which would definitely leave enough time for cutaneous signals to contribute to online control.

Most surfaces encountered during natural object manipulation have frictional properties that cause stick-and-slip when the tangential load is sufficiently large, that is, local breaking of the bond between the surface and even a single fingerprint ridge. However, stick-and-slip events were largely absent in our experiment because we had to use a flat and transparent material to be able to measure skin surface deformations. With natural material, partial slip might therefore occur in a stepwise manner and result in readily measurable accelerations (*Johansson and Westling, 1984*; *Johansson and Westling, 1987*), as opposed to a very smooth progressive wave observed in our experiments. Nevertheless, while the timing might be affected, the patterns of deformation observed in their purest form in this study will generalize in some form to other surfaces and therefore the observations made in our study are likely to hold with natural materials.

There are several reasons why the models' $R^2$ are relatively low. First, the models are linear, whereas the response of tactile afferents to skin deformations is known to be far more complex than accounted for by simple linear relationships (*Dong et al., 2013*; *Saal et al., 2017*). Second, the inputs of the model were obtained from a single point inside the receptive field of the afferent. Yet, we know that human tactile afferents have complex receptive fields with multiple zones of sensitivity that they owe to the branching of the afferent terminals in the skin (*Johansson, 1976*; *Phillips et al., 1992*; *Pruszynski and Johansson, 2014*). A richer stimulus using more stimulation conditions (tangential speeds, normal forces, more orientations) to reduce the effect of the correlations inherently present in the strain pattern might provide sufficient data to identify multiple zones of sensitivity, using the STA approach developed in this study. However, combining our stimulus with another one dedicated to identifying such receptive field topography (*Jarocka et al., 2021*) would be needed to establish a causal relationship between the two. Finally, our setup does not provide the ability to measure all aspects of local skin deformation as discussed above; we were unable to measure deformations along the axis normal to the glass surface. Analyses of strain normal to the contact surface and deformations outside the contact area would require other approaches, for example, finite element analysis.

## Conclusion

Our sense of touch enables us to grasp and manipulate objects with great dexterity. The ability to capture and extract very subtle mechanical phenomena arising continuously during fine manipulation is probably a key to its success. We have demonstrated in this study that subtle skin compressions taking place in the contact area with the object's surface before the slippage provide essential feedback for grasp stability. Given that estimating friction is a complex problem influenced by many factors of the object itself (texture, adhesion, hydrophilic or hydrophobic properties) and by properties of the skin (for instance, stiffness), some of which are constantly changing (e.g., humidity), the nature of this feedback and its independence from friction makes it particularly well suited for grasp stability.

# Materials and methods

## Participants

Sixteen healthy human subjects (seven females; ages 19–24 years) participated in the experiment. Each subject provided written informed consent to the procedures, and the study was approved by the local ethics committee at the host institution (Université catholique de Louvain, Brussels, Belgium). Subjects were seated in a comfortable dentist chair with the forearm slightly pronated and abducted. The forearm was resting on a horizontal cushioned support. The right hand, with the volar side facing the ground, was fixed to a custom-made support (*Figure 1A*), which enabled to lock one finger while keeping the rest of the fingers away in a safe position. The participant's fingers were stabilized by gluing the nail of digit II and III to aluminum bars using cyanoacrylate and connecting the bars rigidly with the custom-made support. Such fixation hindered finger movement during the stimulation but allowed fingertip deformation due to the compliance of the fingerpad.

## Apparatus

The stimulations were applied by a robotic platform based on an industrial robot, as already described in previous studies (*Delhaye et al., 2014*; *Delhaye et al., 2016*; *Barrea et al., 2018*). Briefly, a transparent plate was mounted horizontally on the end effector of an industrial robot (Denso Robotics, Japan) and its movement in three orthogonal directions could be controlled precisely. Two force transducers were mounted, one on each side of the plate, and measured the forces applied to the subject's fingertip (ATI force sensors, ATI Industrial Automation, USA). The measured RMS error of the force measurement was low (ranging 0.01–0.02 N along the tangential axes and 0.03–0.06 N along the vertical axis). The stimulus was applied to the finger where the receptive field of the recorded afferent was located. The relative angle between the stimulus and the finger was around 30°. The transparent plate consisted of smooth glass and was either plain or covered with a hydrophobic coating, RainOff (Arexons, Italy) reducing friction. The robot followed a programmed tangential (horizontal) trajectory with the help of the manufactory position controller. A custom PID controller was developed to servo-control the normal (vertical) force applied to the fingerpad by feeding back the force measurements (average RMS error is 5% during the whole tangential loading, and average peak error is 12% or 0.5 N).

The robot was combined with a custom-made fingerprint imaging apparatus composed of a high-speed (50 fps) and high-resolution (1200 dpi) camera (Mikrotron MC1362, 1280 × 1024 pixels, Mikrotron GmbH, Germany) and a coaxial light source (White LED Backlight, Phlox, France). This optical apparatus enabled imaging of fingerprints in contact with the transparent stimuli to compute strains in the fingertip contact, as described in *Delhaye et al., 2016*.

## Experimental procedures

We used the microneurographic technique (*Vallbo and Hagbarth, 1968*) to record single skin afferent activity ('unit'). An insulated tungsten needle electrode was percutaneously inserted into the right median nerve ~10 cm proximal to the elbow joint (*Figure 1A*). Once the recording electrode was in an intraneural position, it was manipulated in minute steps until single-unit activity clearly stood up from the background noise. To evoke the responses of the afferents, the relevant skin areas of the fingertips were stimulated. Once a single afferent activity was identified and a corresponding receptive field located, the threshold force for the receptor was determined using von

Frey hairs. According to well-known criteria (*Vallbo and Johansson, 1984*), receptors were identified as slowly-adapting (SA) if the discharge was sustained while pressing with a glass probe for at least 2 s; otherwise, they were considered fast-adapting (FA). The type I units have small, easily located receptive fields, whereas type II units are characterized by large and poorly defined receptive fields. Furthermore, SA-II could be distinguished from SA-I by the regularity of their inter-spike intervals, and FA-II units from FA-I by their response to remote light taps. Once the spot with the highest sensitivity within the receptive field had been marked with a permanent marker, the experimental procedure was initiated. The afferent search was voluntarily biased toward type I afferents because of their relatively small receptive field, which makes them more likely to respond precisely to the local strain patterns recorded within the fingertip contact.

While recording impulses from single tactile afferents innervating the right index or middle finger of subjects, the stimulus was moved vertically toward the fingerpad in single trials, made contact, and reached a preset normal force of 4 N ('contact' phase). That normal force was kept constant during the entire trial. After a delay of 1.2 s, which was necessary to induce occlusion and make the contact visible on the camera (*Dzidek et al., 2017*), the surface was moved tangentially (horizontally) with a constant speed (5.5 mm/s) first in one direction (ulnar, radial, distal, or proximal) for 8 mm ('forward' movement) and then in the opposite direction for 12 mm ('backward' movement) (*Figure 1B*). The relatively slow tangential speed is explained by two reasons. First, since our optical system has a limited frame rate (50 Hz), we wanted to have a relatively slow stick-to-slip transition such that it was accurately measured by the imaging system. Second, the normal force servo-control was more effective at low speed. The movement amplitudes were thus large enough to ensure eventually full sliding between the fingertip and the surface in both directions. The surface was then retracted from the finger, and the trial was ended. The procedure was the same for all trials. Each protocol consisted of five repetitions of the four stimulation directions for each of the two friction conditions, totalizing 40 trials (2 frictions × 4 directions × 5 repetitions). For practical reasons, the same sequential order was used for all protocols: all high friction trials (on plain glass) were run first, followed by all low friction trials (on coated glass). Moreover, the direction sequence was always the following: ulnar, distal, radial, and proximal, with all five repetitions made in a row.

Plate position and forces exerted on the finger were sampled at 1 kHz along the three axes (PCIe-1433, National Instruments, and LabVIEW). Fingerprint deformations were monitored through the transparent plate during the tangential movement of the platform, allowing the derivation of surface strains at the fingertip contact. The neural signals were sampled at 12.8 kHz after amplification close to the recording site. The identification of single action potentials was made semiautomatically under visual control (*Edin et al., 1988*). WINSC/WINZOOM software (Umeå University, Sweden) was used for recording and analyzing the neural data. The instantaneous firing rate (as shown in *Figures 2* and *4*) was defined as the inverse of the time interval between two consecutive spikes for the interval duration. This rate was resampled at 1 kHz to obtain convenient time series for data analysis.

## Image processing

For each trial, strains in the contact area were computed from the images as described in *Delhaye et al., 2016* (*Figure 1C*). Briefly, the contact region was first extracted semiautomatically from each image of the sequence. Second, equally spaced features were sampled in the contact area of the initial frame and then tracked from frame to frame to measure the displacement field in the contact using the optical flow technique (*Lucas and Kanade, 1981*) implemented in the OpenCV online computer vision toolbox (*Bradski, 2008*). Third, the tracked features were triangulated (Delaunay triangulation), and Green-Lagrange strains were computed for each triangle by calculating the gradient of the displacement field. This operation yielded a 2-by-2 symmetric strain matrix for each triangle and each pair of consecutive frames. Axial strains, that is, the diagonal elements of the strain matrix, were denoted $e_{xx}$ and $e_{yy}$ and shear strain (off-diagonal) was denoted $e_{xy}$. The x-axis was aligned to the radial-ulnar orientation (*Figure 1B*). Strains were filtered, first spatially (using Smooth Triangulated mesh from Matlab FEX, https://mathworks.com/matlabcentral/fileexchange/26710-smooth-triangulated-mesh), and then temporally (median filtering over three strain values). By using the term 'strain' throughout the article, we refer to the elements of the strain tensor computed between two consecutive images and expressed in percent per second, that is, strain rates.

The principal strain components denoted $e_1$ and $e_2$ were then obtained by an eigenvalue/eigenvector decomposition of the strain matrix. The principal strains $e_1$ and $e_2$ correspond to the maximum

compressive and tensile strains at the location where the strain tensor is measured. The eigenvalue decomposition is equivalent to a rotation of the reference frame so that the shear strain is canceled and only axial strains remain, which thus corresponds to the maximum local compression and/or dilation. Therefore, the principal strains are not necessarily aligned with the axes of the $x$-$y$ reference frame defined above and their orientation is potentially different for each local strain tensor. Therefore, the matrix components of the local strain tensor $e$ are solely given by the axial and shear strains along the $x$-$y$ plane ($e_{xx}$, $e_{yy}$, and $e_{xy}$). Besides, given the high stiffness of the outer layer of the skin (*Wang and Hayward, 2007*), it is reasonable to assume that the area of local patches of skin is conserved after deformation. Under this hypothesis, it can be shown that $e_1$ and $e_2$ have opposite signs. This means that if one of the principal strain is compressive, the other one is dilative and vice versa. Moreover, we follow the convention to sort the eigenvalues by ascending order. Therefore, $e_1$ will always be smaller than $e_2$. Under the hypothesis of area conservation, this implies that $e_1$ will be negative and thus compressive, whereas $e_2$ will be positive and thus tensile. This hypothesis has been verified in practice on the data.

To assess the strains taking place at each given point of the contact area across different trials, we interpolated the strains on an arbitrary rectangular grid, the 'strain grid'. First, an easily identifiable feature of the fingerprint located near the center of contact was manually spotted on the first image of each trial and used to precisely align all trials. Given the precise repeatability of the trials, the spotted feature only moved by a few pixels from trial to trial. Then, the contact area was centered on the grid using its geometric center on the first image. The feature coordinates on the first image were then used to interpolate the strains over the entire trial on the given grid using MAT-LAB's *scatteredInterpolant* function. The 'strain grid' spacing was set to 10 pixels (~200 um) and was composed of 120-by-90 elements (1200 × 900 pixels).

The local fingerprint orientation was also estimated as described previously (*Delhaye et al., 2016*) by computing the image gray-level gradients (*Bazen and Gerez, 2000*) over a 60-by-60 pixels window.

## Data analyses

### Receptive field location and size

The spot of highest sensitivity in the receptive field was marked on the finger, and a close view picture of the finger was taken at the end of the experiment. For the afferents inside or on the border of the contact area, the location and local fingerprint pattern were visually identified on the picture and matched with the pattern recorded using the bottom view camera (see insets in *Figure 2B, D* for instance), to obtain the coordinates of the receptive field center in the image recordings. The receptive field of a type I afferent was defined as a circle around this central coordinate with a radius of 150 pixels (~3 mm, *Johansson and Vallbo, 1979*).

### Phases of tangential loading

Forward and backward movements of the glass surface loading the fingerpad were divided into three sequential phases. The *onset* started with the tangential (horizontal) movement of the platform and lasted 100 ms. This value was set arbitrarily as short as possible but also ensuring that the transient response due to the start of the motion observed in some afferents did not affect the partial slip phase. The *partial slip* phase followed and lasted until the tangential force plateau was reached, namely, when the tangential force derivative returned close to zeros. At this point, the finger is *fully slipping*. The plateau phase ended when the plate movement stopped. Afferent responses were related to the local strains only during the partial slip phase.

### Afferents' recording quality

The quality of the electrophysiological recordings was assessed by computing the spikes' signal-to-noise ratio (SNR). The SNR was defined as the ratio of the mean peak-to-peak amplitude of spikes to the mean of peak-to-peak noise (measured from a 1-ms-long window before each spike). The noise level was subtracted from the spike signal before calculating the ratio. The mean SNR across all units is 4.3 (range 2.1–7.9).

## Afferents' effective sampling frequency

We checked that the frequency content of the afferent response matched the optical recording system by estimating their 'effective sampling frequency' (*Dawdy and Matalas, 1964*; *Dimitriou and Edin, 2008*). We found that, indeed, the stimulus did not generate effective information above 50 Hz; across afferents, median effective sampling frequency was 15 Hz ($Q_{1-3}$ = 3.5–38.5 Hz). Such low-frequency content is explained by the slow nature of the stimulus.

## Spike-triggered average

 The STA matrices were computed only during the partial slip phase, and data from all conditions and repetitions were combined. The spikes were first binned into 20 ms bins to match the image/strain frame rates (given the afferents' frequency content, such time-bin resolution is adequate). The strains consisted of a matrix of 90-by-120 elements covering the contact area for each time bin. First, strain frames from all the corresponding bins containing at least one spike were selected, later referred to as the 'spike selection'. Frames corresponding to bins containing more than one spike were repeated (n times for n spikes per bin). The median number of spike used to build the STA was 373. Second, the same number of strain frames was randomly chosen from bins that did not contain spikes (referred to as 'no-spike selection'). Then, the average of the no-spike selection was subtracted from the average of the spike selection. The result was a 90-by-120 matrix defined as the STA. The sum of the standard deviation of the two selections was also obtained. The STA elements that were lower than four times the sum of the standard deviations were set to 'NaN' (not a number). The STA elements that were not inside the contact area for at least 50% of the time were also set to 'NaN'. It is possible to compute an STA for any strain component. In *Figure 5*, we chose to compute the STA for the strain norm, which is defined as the square root of the sum of the square of all elements of the strain tensor ($||e|| = \sqrt{e_{xx}^2 + e_{yy}^2 + 2 \cdot e_{xy}^2}$). The strain norm has no physical meaning but provides an estimate of the intensity of the deformation at any point and is rotation invariant, and therefore not dependent on the choice of a given reference frame. When expressed as the strain norm, we expect the stimulus to take the shape of an annulus of deformation moving from the periphery to the center. Therefore, the STA is expected to tend toward the shape of an annulus.

## Linear regression model and cross-validation

For each element of the 90-by-120 'strain grid' covering the contact area, we fitted a multiple regression model to predict the afferent firing rate. Only the time bins in the partial slip phase were used, and only the elements of the matrix that were inside the contact area for more than 50% of the time were used. We used regression with six distinct predictors and an intercept. Each of the three strain components ($e_{xx}$, $e_{yy}$, and $e_{xy}$) were half-wave rectified using both the positive (related to stretch and positive shear, $e_{xx}^+$, $e_{yy}^+$, and $e_{xy}^+$) and negative part (related to compression and negative shear, $e_{xx}^-$, $e_{yy}^-$, and $e_{xy}^-$). The firing rate was obtained as follows. The vector of the spike times, sampled at 1 kHz and made of 1's at the time of a spike and 0's elsewhere, was first convolved by a Gaussian window (using MATLAB's *gausswin* function, with 480 points and an alpha value of 6, normed) and multiplied by 1000. The windows' width was empirically found to maximize the prediction performance across afferents (suggesting that our stimulus frequency content is centered around 4 Hz). Second, the result was resampled at the image frame rate (50 Hz). The result was defined as the firing rate and used to fit our models. We used three cross-validation schemes. In the first, the data was split in two (twofold CV), with the high friction trials in one fold and the low friction trials in the other fold. In the second twofold scheme, the first fold consisted of the data during the forward movement and the second fold the backward movement. In the third, fourfold, scheme, each fold consisted of all trials in a given direction (ulnar [U], distal [D], radial [R], or proximal [P]). For all schemes, the model was fitted on the data that excluded one fold and tested for prediction on that fold. The procedure was repeated for each fold such that a prediction was made for all data points. The $R^2$ was obtained for each scheme by combining prediction from each fold.

## Component selection and rotation

The same cross-validation procedure was followed to fit linear regressions with different predictors. All models included an intercept. For that procedure, only FA-I afferents inside the contact area

were used, and the strains coming from a single location were used: the location inside the afferent receptive field yielding the maximal performance for the full model (with six predictors). The strain tensor was recomputed in a rotated reference frame following this transformation:

$$\begin{bmatrix} e'_{xx} & e'_{xy} \\ e'_{xy} & e'_{yy} \end{bmatrix} = \begin{bmatrix} \cos\theta & \sin\theta \\ -\sin\theta & \cos\theta \end{bmatrix} \begin{bmatrix} e_{xx} & e_{xy} \\ e_{xy} & e_{yy} \end{bmatrix} \begin{bmatrix} \cos\theta & -\sin\theta \\ \sin\theta & \cos\theta \end{bmatrix}$$

The reference frame was rotated from 0 to 90° with steps of 5°. When the frame is rotated by 90°, the rotated x-axis corresponds to the initial y-axis (*Figure 6B*). Four different models were tested using this procedure, with a single predictor each, using half-wave rectified strain along the two axial components.

### Statistical analyses

All statistical analyses were performed in MATLAB using the functions *corr* (for Pearson correlation), *ttest* (for paired t-tests), and *ranova* (for one-way analysis of variance with repeated measures). For the *ranova*, the normality and sphericity were verified and the Greenhouse–Geisser adjustment was applied if needed. The test performed, degrees of freedom, T or F statistic is always mentioned with the p-value.

## Acknowledgements

We thank Hannes Saal, Vincent Hayward, and the members of the DEX group at UCLouvain for useful comments on a previous version of the manuscript. We also thank Julien Lambert for invaluable logistical support. This work was supported by a grant from the European Space Agency, Prodex (BELSPO, Belgian Federal Government), and the Swedish Research Council (VR 2016-01635). BPD is a postdoctoral researcher of the Fonds de la Recherche Scientifique – FNRS (Belgium).

## Additional information

### Funding

| Funder | Grant reference number | Author |
|---|---|---|
| European Space Agency | | Jean-Louis Thonnard<br>Philippe Lefèvre |
| PRODEX | BELSPO | Jean-Louis Thonnard<br>Philippe Lefèvre |
| Swedish Research Council | VR 2016-01635 | Benoni Edin |
| Fonds De La Recherche Scientifique - FNRS | FNRS | Benoit P Delhaye |

The funders had no role in study design, data collection and interpretation, or the decision to submit the work for publication.

### Author contributions

Benoit P Delhaye, Conceptualization, Data curation, Software, Formal analysis, Validation, Investigation, Visualization, Methodology, Writing - original draft, Writing - review and editing; Ewa Jarocka, Allan Barrea, Data curation, Investigation, Writing - review and editing; Jean-Louis Thonnard, Philippe Lefèvre, Conceptualization, Supervision, Funding acquisition, Writing - review and editing; Benoni Edin, Conceptualization, Supervision, Investigation, Writing - review and editing

### Author ORCIDs

Benoit P Delhaye https://orcid.org/0000-0003-3974-7921
Allan Barrea http://orcid.org/0000-0002-1094-4596
Philippe Lefèvre http://orcid.org/0000-0003-2032-3635

## Ethics

Human subjects: Each subject provided written informed consent to the procedures, and the study was approved by the local ethics committee at the host institution (Institute of Neuroscience, Université catholique de Louvain, Brussels, Belgium).

## Decision letter and Author response

Decision letter https://doi.org/10.7554/eLife.64679.sa1
Author response https://doi.org/10.7554/eLife.64679.sa2

# Additional files

## Supplementary files

• Transparent reporting form

## Data availability

All the data used to create the figures in the manuscript are available for download following this Zenodo link: https://zenodo.org/record/4818439.

The following dataset was generated:

| Author(s) | Year | Dataset title | Dataset URL | Database and Identifier |
|---|---|---|---|---|
| Delhaye BP, Jarocka E, Barrea A, Thonnard J, Edin B, Lefèvre P | 2021 | High-resolution imaging of skin deformation shows that afferents from human fingertips signal slip onset | https://zenodo.org/record/4818439 | Zenodo, 4818439 |

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
