## [Decision Letter]

**Acceptance summary:**

Delhaye et al., study the role of local tangential skin strain for tactile neuronal encoding in humans, variables that were not accessed by classic studies. They visualized the fingerprint when moving across a smooth surface together with extracellular recordings of primary afferents with a receptive field on or close to the fingerprint. Focusing on the period, in which the fingertip partially loses contact with the surface and starts to move (partial slip), they found that fast-adapting primary afferents type 1 (FA1) consistently respond to local strain, with predominance of compression over stretch, while the slowly adapting type 1 (SA1) do not. The FA1 responses during partial slip show selectivity to directional/orientation of the compression wave with some initial insights about the contribution of papillary ridges at the site of the neuron's receptive field.

**Decision letter after peer review:**

Thank you for submitting your article "High-resolution imaging of skin deformation shows that afferents from human fingertips signal slip onset" for consideration by *eLife*. Your article has been reviewed by 2 peer reviewers, including Cornelius Schwarz as the Reviewing Editor and Reviewer #1, and the evaluation has been overseen by Richard Ivry as the Senior Editor. The following individual involved in review of your submission has agreed to reveal their identity: Rochelle Ackerley (Reviewer #2).

The reviewers have discussed the reviews with one another and the Reviewing Editor has drafted this decision to help you prepare a revised submission.

Summary:

Delhaye et al., study the role of tangential skin deformation for tactile encoding in humans, variables that were not accessed by classic studies. They use visualizing the fingerprint when moving across a smooth surface together with extracellular recordings of primary afferents with a receptive field on or close to the fingerprint. 'Fast-adapting type 1, one out four classes of human primary afferents, are shown to respond to strain, rather than stretch when the fingerprint's adherence to the surfaces goes from a fixed state to a partial attachment to full slippage.

Essential revisions:

The reviews both see considerable merit in your work. They agree that it contains considerable advances in the field of human tactile coding by combining, for the first time, the visualization of skin strain patterns and microneurography. They state the results on FA1 will be important to students of perception as well prehension. Both generally support future publication. However, they state major comments as well, which I will summarize below. I consider the major points all important enough to expect you to address them either with new analyses or through improvement your treatment of them in the text.

1. The most important criticism was raised unanimously by the reviews. It should be addressed by adequate new analyses. It states that while making excellent points about FA1 afferents, there is a deficiency in covering SA1 primary afferents. (The other two classes are considered to be not sufficiently sampled and should be clearly labeled as circumstantial findings). This deficiency was considered most remarkable as SA1 do fire vigorously to presentation of the stimuli. The authors argue that the unexplained SA1 activity was mainly due to vertical skin deformation that they cannot monitor. In apparent contrast, however, the experiment was designed to abolish or minimize normal, i.e. vertical forces. The authors need to clarify this issue. They should explain what the purpose of robotic normal force control was, and importantly, in how far it worked (or did not work). What is the precision of their normal force measurement? Is the measurement suited to relate spike trains to time series of measured normal force? If yes, the reviewers recommend to use those measurements. In the discussion the authors need to clarify how their speculation about vertical forces activating SA1, relates to the robotic normal force control and measurements and Figure S1.

The reviewers feel that figure S1 is a good start, but more detailed analysis in this direction would be helpful. A possible temporal relationship of tangential deformations and SA1 firing is not sufficiently analyzed. The analyses used (STA and regression) are not systematically focused to bring out temporal relationships between SA1 and skin deformation. The implemented STA analysis seems to consider just one time-bin of 20 ms length for spiking and strain (the camera frame rate), while, in unexplained discrepancy, the regression analysis uses convolution of the spike train with a Gaussian window of 480 ms length – strongly diluting the temporal relationship of spike and strain parameters. The authors need to explain the rationale using the two extreme temporal settings in the two analyses. And they should apply a systematic approach to analyze temporal relationships of spikes and strain maps.

Maybe the authors even have runs/sessions without normal force control? If yes, these could be analyzed and presented.

Toward SA1 coding the reviewers specifically recommended to study/analyze the following:

Is it feasible that SA1 spiking dynamics are different for different directions of skin deformations (indentation vs. horizontal), e.g. could SA1 be slow for vertical but fast for horizontal stimulation, or vice versa, etc.?

The authors should go beyond their present attempts (mainly Figure S1), and build a model that is focused to find out the temporal relationship of strain/stretch/shear as well as normal forces and SA1 spikes.

Could the STA and regression analysis be helped by fitting responses to past skin deformations at longer delays than 20 ms (STA) or 12 ms (Figure S1)?

The authors use 'correlations' of deformation and spikes as basis for their arguments. Maybe it would be more appropriate to calculate delayed correlation, i.e. e.g cross correlation patterns to help to explain SA1 responses to the strain dynamics?

A further observation was that SA1 code for movement without slipping (in apparent contrast to FA1). A related phenomenon observed in the firing rate traces is a conspicuous silencing of SAI firing during movement (especially in the low friction condition). Is it feasible that SA1 signal touch well – albeit through the absence of firing?

It is interesting to note that the only afferents that have any response during movement, but not slipping, are SA units. This would be expected, but it is interesting that the non-slip moving phase show overall less firing than in the initial stationary load phase. Do the authors think that this could be an effect of the general decrease in firing frequency (i.e. adaptation) from SA units (with a re-increase in firing from slip) or is it more specific and somehow related to a difference in fingertip forces? This second idea about a real difference in firing between the stationary and moving-without-slip phases could imply that SA units do encode such specific aspects of touch, which FAs do not. From Suppl. Figure 2, it certainly seems that there are such differences.

2. Another unanimous point of the reviewers was the perceived insufficiency of the analysis of preferred direction. First of all, the definition (line 167) was not clear. Was it medial/lateral/radial/ulnar or in degrees (what does 0 deg refer to)?. The authors are recommended to calculate a standard directionality index (e.g. (pref-unpref)/(pref+unpref), vector sum, etc.). Further, the reviewers see the need to differentiate preference for 'directionality' from that for 'orientation'. In addition, it is recommended to clarify the following parameters related to the issue of directionality/orientation tuning:

a. Receptive field location

b. Orientation of papillary ridges inside the RF

c. Directionality relative to fingertip or papillary ridges?

In 8 of 13 FA1s the direction preference was not detailed. Was it like the other units or did they have no preference? Figure 3C and legend should be improved to clarify these questions.

3. From figure 4 it appears that responses to the same movement direction performed as 'forward' and 'backward' are very different. The reviewers felt that this phenomenon deserves a proper treatment. The authors are asked to clarify how consistent the phenomenon is, and whether there is a possible explanation in terms of different contexts that leads to it? It is considered worth to report whether SA1 show the same.

In Figure 4 please label the top and bottom series of strain maps – what exactly do they represent? High/low friction? The touch does not cover the receptive field in the bottom maps, what does that mean, which effect may it have on firing?

4. The reviews also noted that STA analysis was only done with ||e||. As a main conclusion of the paper is about responses to strain versus stress, shouldn't STA be performed with *e_xx_*, *e_yy_*, *e_xy_* to differentiate those parameters? In this respect the question was raised why the authors focus on the annular pattern. Given the annular form of the partial slip and the high-correlation of deformations across the fingertip such a pattern is trivially expected (especially using ||e||). The authors are recommended to use the mentioned more specific parameters *e_xx_*, *e_yy_*, *e_xy_*, and focus first on the RF, and second on deformations elsewhere (maybe using some decorrelation techniques).

5. It was further suggested to attempt using GLMs or similar to capture non-linearities between strain and spikes. The rigid linear model will fail to capture expected non-linear relationships between spike rates and other parameters.

6. Both reviewers found the part of the discussion stimulating, in which the authors touch upon orientation selectivity (line 351-354). It seems an interesting point to consider the differences between slips and edges – both in forces and neural information. Do the authors have evidence to think the brain could tell the difference between a slip and an edge? Alternatively, the speculation starts to explain orientation selectivity of primate tactile neurons in terms of prehension (rather than perception of edge orientation, as is typically assumed). This point is considered to be an outcome of their novel measurements. It should be worked out and presented more prominently.

---

## [Author Response]

Essential revisions:The reviews both see considerable merit in your work. They agree that it contains considerable advances in the field of human tactile coding by combining, for the first time, the visualization of skin strain patterns and microneurography. They state the results on FA1 will be important to students of perception as well prehension. Both generally support future publication. However, they state major comments as well, which I will summarize below. I consider the major points all important enough to expect you to address them either with new analyses or through improvement your treatment of them in the text.

Thank you for this very deep and thorough review. We greatly appreciate the reviewers’ effort to help us to improve the paper. We have tried to address each comment adequately and this resulted in significant revisions of the manuscript. The manuscript has been much improved thanks to the reviewers’ feedback.

Overall, we believe that several major comments made by the reviewers at least partly originate from a failure on our side to make a clear distinction between “global” and “local” aspects of the stimulation and the measurements used in our study. On the one hand, the afferent responses were briefly studied with respect to “global” parameters, such as the different stimulus phases (tangential movement: onset, partial slip and plateau; Figure 3), the stimulus direction (i.e., ulnar, distal, radial and proximal; Suppl. Figure 1) or the force measurements (Suppl. Figure 2). This short analysis was aimed at providing a general context and relating this study to previous works covering those aspects in more details (e.g. Birznieks et al., 2001 JNeuroSci, Johansson and Birznieks 2004 Nat NeuroSci, Birznieks et al. 2009 JNeuroSci, Khamis et al. 2015 JNeuroPhysiol). On the other hand, however, our main focus, which is the main originality of our work, was to relate the afferent responses to “local” skin deformations which occur within the afferent’s receptive field and which can be measured with our optical system (Figure 2, 4, 5, 6 and Supp Figure 3, 4, 5 and 6). We have made numerous changes at several places in the manuscript (detailed below) to make this crucial difference clearer. Moreover, we added several important details in the Methods to better explain our choices. Finally, to address the reviewers’ major concerns, we provide new analyses that are reported in the Methods and also in the *new* supplementary figures 3 and 6.

Note that the order of the Supplementary Figures 1 and 2 has been flipped to accommodate their order of appearance in the main text.

1. The most important criticism was raised unanimously by the reviews. It should be addressed by adequate new analyses. It states that while making excellent points about FA1 afferents, there is a deficiency in covering SA1 primary afferents. (The other two classes are considered to be not sufficiently sampled and should be clearly labeled as circumstantial findings). This deficiency was considered most remarkable as SA1 do fire vigorously to presentation of the stimuli.

Yes, indeed, the SA-I afferents do fire vigorously to the tangential loading. In Supplementary figure 2 (former Supplementary figure 1), we show that these vigorous responses are at least partially related to the fluctuations of the tangential forces, a global measurement. Nevertheless, we failed to find any correlate of those discharges in our “local” measurements of the skin strain. The lack of such relationship may suggest that the SA-I afferents encode other, non-measured aspects of the skin deformation. This is the only conclusion that we can draw with respect to the SA-I afferent responses to the “local” measured skin deformation which we aimed to explore in this study.

Following this comment, in the Discussion, we re-wrote the paragraph about the lack of relationship between SA-I responses and local skin strain hypothesizing two possible aspects of the deformations that might be encoded by these afferents.

“Our methods generate high-resolution representations of the distribution of surface strain in the contact area of the fingertip (i.e. 2D, x and y axes) but do not allow measurements of the distribution of the deformations perpendicular to the fingertip surface inside the contact area (i.e. along the z-axis), nor outside the contact area. […] When considering the subdermal location of SA-II and FA-II afferents terminals, it is not surprising to observe a poor coupling between their responses and the measured surface strains.”

We would also like to stress that due to the complexity of the microneurography experiment, i.e. finding an afferent with a receptive field inside the contact area and maintaining the recording sufficiently long to finish a full protocol, we were able to obtain data only from two SA-I afferents fulfilling these criteria (six in total, but two inside the contact area). This is another reason why the main focus of the analyses is on the responses from FA-I afferents.

The authors argue that the unexplained SA1 activity was mainly due to vertical skin deformation that they cannot monitor. In apparent contrast, however, the experiment was designed to abolish or minimize normal, i.e. vertical forces. The authors need to clarify this issue. They should explain what the purpose of robotic normal force control was, and importantly, in how far it worked (or did not work). What is the precision of their normal force measurement? Is the measurement suited to relate spike trains to time series of measured normal force? If yes, the reviewers recommend to use those measurements. In the discussion the authors need to clarify how their speculation about vertical forces activating SA1, relates to the robotic normal force control and measurements and Figure S1.

We believe that there was a confusion in the Discussion about the SA-I afferents with respect to their response to global parameters, i.e. normal and tangential force (measured), and to the local skin deformations, normal (not measured) and tangential (measured) to the surface. What is proposed in the Discussion is that SA-I afferents may be sensitive to the strains normal to the skin surface (not measured) rather than to the tangential strains we measured, which failed to explain SA-I firing rates. In the revised manuscript under Limitations of the study we have described this in detail and also throughout the manuscript we have now attempted to make the distinction between global and local parameters very clear (see response to comment #1).

The robot controller was intended to maintain the normal force constant and was successful with a RMS error of only 5% (average peak error is 12%) and our force measurements were precise. The normal force control is necessary because if the surface is maintained at a constant vertical position, the lateral movement and increase of tangential force generate an increase in normal force (i.e. “poynting effect”). In order to dissociate the two forces, it is therefore necessary to lower the vertical position as the tangential movement is launched.

We have modified the Methods and the last paragraph of the Results to accommodate this information. Given our sampling frequency (1 kHz), the measurements are indeed well suited to relate the time series to the spike trains.

Methods:

“A custom PID controller was developed to servo-control the normal (vertical) force applied to the fingerpad by feeding back the force measurements (average RMS error is 5% during the whole tangential loading and average peak error is 12% or 0.5N).”

Results:

“Finally, we asked how much the response of all types of afferents was related to the “global” external 3D force vector. […] Their firing rates were however not related to the small vertical (normal) force fluctuations.”

The reviewers feel that figure S1 is a good start, but more detailed analysis in this direction would be helpful. A possible temporal relationship of tangential deformations and SA1 firing is not sufficiently analyzed. The analyses used (STA and regression) are not systematically focused to bring out temporal relationships between SA1 and skin deformation. The implemented STA analysis seems to consider just one time-bin of 20 ms length for spiking and strain (the camera frame rate), while, in unexplained discrepancy, the regression analysis uses convolution of the spike train with a Gaussian window of 480 ms length – strongly diluting the temporal relationship of spike and strain parameters. The authors need to explain the rationale using the two extreme temporal settings in the two analyses. And they should apply a systematic approach to analyze temporal relationships of spikes and strain maps.

First, it is important to stress that we chose a relatively slow tangential speed for the robot (5 mm/s), for two limiting reasons. 1) so that the stick to slip transition could be accurately measured at the frame rate (50 fps) and 2) so that the normal force control could be effective.

Second, the image system time resolution is indeed 20 ms (50 Hz), which is why our STA analyses were performed at that resolution.

Third, we checked that the afferent response frequency content matched the optical recording system by estimating their “effective sampling frequency” (see references below). We found that indeed, the stimulus did not generate effective information above 50 Hz in the afferent responses (median effective frequency 15 Hz, quartiles 3.5 – 38.5 Hz). The STA analysis with a 20 ms time-bin resolution is therefore well suited to capture all the frequency spectra of the afferent responses.

Finally, the Gaussian window width (effective cutoff 4 Hz) used to smooth the spike trains was found such that it empirically maximized the prediction performance across afferents. Such a low cutoff is not surprising given the elements above, and suggests that our stimulus actual frequency content was centered at 4 Hz.

We have amended the Methods to accommodate the above-mentioned information.

“Afferents’ effective sampling frequency. […] We found that indeed, the stimulus did not generate effective information above 50 Hz; across afferents, median effective sampling frequency was 15 Hz (Q1-3 = 3.5 – 38.5 Hz). Such low frequency content is explained by the slow nature of the stimulus.”

“The vector of the spike times, sampled at 1 kHz and made of 1’s at the time of a spike and 0’s elsewhere, was first convolved by a Gaussian window (using MATLAB’s gausswin function, with 480 points and an alpha value of 6, normed) and multiplied by 1000. The windows width was empirically found to maximize the prediction performance across afferents (suggesting that our stimulus frequency content is centered around 4 Hz).”

Dawdy DR and Matalas NC (1964). Statistical and probability analysis of hydrologic data, part III: Analysis of variance, covariance and time series. In Handbook of Applied Hydrology, a Compendium of Water-Resources Technology, ed. Chow VT, pp. 68–90. McGraw-Hill Co., New York.

Dimitriou M, Edin BB (2008) Discharges in human muscle spindle afferents during a key-pressing task. J Physiol 586:5455–5470;

Maybe the authors even have runs/sessions without normal force control? If yes, these could be analyzed and presented.

No, we did not run such sessions.

Toward SA1 coding the reviewers specifically recommended to study/analyze the following:Is it feasible that SA1 spiking dynamics are different for different directions of skin deformations (indentation vs. horizontal), e.g. could SA1 be slow for vertical but fast for horizontal stimulation, or vice versa, etc.?

Unfortunately, we do not have the data for such analysis, nor the methods to obtain such data (our imaging technique does not allow to study perpendicular skin deformation). We were specifically interested in the tangential loading of the fingerpad in the context of grasp stability thus we did not strive to study skin indentation.

The authors should go beyond their present attempts (mainly Figure S1), and build a model that is focused to find out the temporal relationship of strain/stretch/shear as well as normal forces and SA1 spikes.

Our local measurements provide a mechanistic description of the deformations in the receptive field of an afferent, while the “global” force measurements are a consequence of the global deformation of the fingertip. We believe that combining “global” and “local” parameters into a single model will not help to explain the relationship between the skin afferent responses and local skin deformations. Adding the forces to the model might increase the amount of variance that can be explained by the model but will not inform us about the nature of the local skin deformations that generated spiking activity (the aim of our study).

Could the STA and regression analysis be helped by fitting responses to past skin deformations at longer delays than 20 ms (STA) or 12 ms (Figure S1)?The authors use 'correlations' of deformation and spikes as basis for their arguments. Maybe it would be more appropriate to calculate delayed correlation, i.e. e.g cross correlation patterns to help to explain SA1 responses to the strain dynamics?

Thank you for the suggestion. Unfortunately, given the slow nature of our stimuli (see above), the frequency content of the response, the short pathway and the firing properties of the primary afferent and the image sampling frequency, we do not believe that this approach is appropriate.

A further observation was that SA1 code for movement without slipping (in apparent contrast to FA1). A related phenomenon observed in the firing rate traces is a conspicuous silencing of SAI firing during movement (especially in the low friction condition). Is it feasible that SA1 signal touch well – albeit through the absence of firing?

We are not sure that we understood the reviewers first point and to which figure it referred. As mentioned above and now in the Discussion paragraph about the SA-I afferents responses, the pressure distribution can change during tangential loading and might be strongly reduced under certain circumstances for afferents close to the border of the contact, which might explain the silencing. Again, we did not measure the pressure distribution so this is only speculation. Moreover, the rolling of the finger could make the afferents lose contact with the glass plate. However, following deep data inspection, we did not find any systematic silencing when the contact was lost.

It is interesting to note that the only afferents that have any response during movement, but not slipping, are SA units. This would be expected, but it is interesting that the non-slip moving phase show overall less firing than in the initial stationary load phase. Do the authors think that this could be an effect of the general decrease in firing frequency (i.e. adaptation) from SA units (with a re-increase in firing from slip) or is it more specific and somehow related to a difference in fingertip forces? This second idea about a real difference in firing between the stationary and moving-without-slip phases could imply that SA units do encode such specific aspects of touch, which FAs do not. From Suppl. Figure 2, it certainly seems that there are such differences.

We are not sure how to understand this comment, mainly because there is no “non-slip moving” phase. The main focus of our analyses was on the afferent responses during tangential movement of the plate against the fingerpad i.e., from the moment the fingerpad started slipping until it was in a full slip and the three moving phases were identified as the onset, partial slip and plateau. We monitored skin strains during a transition from a fully stuck contact to a fully slipping contact and tried to relate them to the afferents’ responses. We do not report any results (just display the firing rates from whole trials on the Figure 2 and Supplementary figure 1) from the periods before the tangential movement started or after it fully slipped.

2. Another unanimous point of the reviewers was the perceived insufficiency of the analysis of preferred direction. First of all, the definition (line 167) was not clear. Was it medial/lateral/radial/ulnar or in degrees (what does 0 deg refer to)?. The authors are recommended to calculate a standard directionality index (e.g. (pref-unpref)/(pref+unpref), vector sum, etc.). Further, the reviewers see the need to differentiate preference for 'directionality' from that for 'orientation'.

We believe that this comment is also related to the confusion between global and local direction/orientation preference. On the one hand, we describe a “global” preferred direction related to the robot movement (P/U/R/D). On the other hand, we show that the afferents preferentially encoded “local” compressive strain rates along a given orientation (in degrees).

We have made the following changes to avoid this ambiguity:

In the paragraph related to Figure 3, we gave more details about how we defined the “global” preferred direction, that is in relation with the plate movement and which can only be one of the four directions (ulnar, distal radial or proximal). We changed the Figure 3B to remove angles in degrees, and report direction with respect to the preferred (North) as pointing to cardinal direction (East, South and West). When possible, we also added a word “global” to the preferred directions to avoid confusion, i.e. “preferred global directions”.

“First, we describe how the tactile afferent responded with respect to “global” stimulus parameters such as the movement phase or direction. […] While the firing rates were strongly correlated across friction levels (Figure 3E; correlation r = 0.95, n = 176, p < 0.001), the firing rates across the whole population tended to be slightly lower for low friction (paired t-test, t(171) = 7.57, p < 0.001) and the ratio of the mean firing rate during low and high friction condition was 0.82 ± 0.55 (mean ± standard deviation).”

In the paragraph related to Figure 6, we added two sentences to clarify the concept of local strain orientation preference. We also always use “preferred strain orientation” to avoid confusion.

“It is important to avoid the confusion between the units’ preferred direction described in Figure 3, which relates to the robot movement direction and the maximal firing rate of the afferent, and the strain orientation preference described here, which is related to the orientation of the local deformation in the afferent receptive field.”

Finally, we realized that we had been using the terms “orientation” and “direction” interchangeably at several places in the text, which is very unfortunate given the importance of the distinction between them. We verified that all instances of the use of those words were corrected.

In addition, it is recommended to clarify the following parameters related to the issue of directionality/orientation tuning:a. Receptive field locationb. Orientation of papillary ridges inside the RFc. Directionality relative to fingertip or papillary ridges?

Thank you for this important point. The receptive field location for all neurons is shown in Supp Figure 1. We now also documented the orientation of the papillary ridges for the six FA-I afferents that we could study in details with respect to the strain in a new Supp. Figure 6, which shows the strain orientation preference (obtained from Supp Figure 5) vs the orientation across the fingerprint ridges around their RFs. We also added the local fingerprint pattern for clarity and transparency.

In 8 of 13 FA1s the direction preference was not detailed. Was it like the other units or did they have no preference? Figure 3C and legend should be improved to clarify these questions.

The movement direction preference (Figure 3) was computed for all afferents, and all of them are shown in Figure 3C.

The strain orientation preference was computed only for the afferents for which we had strain data, i.e. for the 6 afferents with their receptive fields inside the contact area. We clarified this point in the paragraph related to the “Aspects of the skin strain rates encoded by the FA-I afferents”.

“Having demonstrated that the recorded FA-I afferents respond to local strain patterns, we then sought to uncover what aspects of the strains were responsible for these responses. […] The analysis was performed only on the FA-I afferents for which we had optical measurements of the strains, that is, those having their receptive fields inside the contact area (n = 6, all shown in the top line of Figure 5A).”

3. From figure 4 it appears that responses to the same movement direction performed as 'forward' and 'backward' are very different. The reviewers felt that this phenomenon deserves a proper treatment. The authors are asked to clarify how consistent the phenomenon is, and whether there is a possible explanation in terms of different contexts that leads to it? It is considered worth to report whether SA1 show the same.

We believe that this comment might stem from a misunderstanding of the figure layout. Forward movement (Figure 4A left), should be compared with backward in the same direction (Figure 4B, right).

As shown in Figure 3D, all the afferents had very similar firing rates in the forward and the backward movements. The afferent shown in Figure 4 follows the same trend: the firing rates elicited in the proximal direction (Figure 4A left and Figure 4B right) are very similar. The timing is different because the timing of the strain wave is different.

One difference is the short burst elicited at the movement onset in the distal backward condition. Unfortunately, the local strains do not explain that burst. Such burst at the very onset of the movement only occurred in a small fraction of the afferents (cfr Figure 3A). We added a sentence to clarify this point.

“Note that a short burst was elicited at the onset of the movement in the distal direction in the backward case (Figure 4A, right). Such transient burst cannot be explained by our strain measurements and occurred in a small fraction of the trials and only in few afferents (Figure 3A).”

In Figure 4 please label the top and bottom series of strain maps – what exactly do they represent? High/low friction? The touch does not cover the receptive field in the bottom maps, what does that mean, which effect may it have on firing?

The two rows show the maps for the two friction conditions. We added labels. And yes, due to rolling, part of the RF could lose contact with the plate. This is part of the unexplained variance of our models. We mentioned this element at the end of the first paragraph of the “FA-I afferents respond to local skin strain rates” section.

“Also note that part of the unit receptive field lost contact during the partial slip phase in the high friction case.”

4. The reviews also noted that STA analysis was only done with ||e||. As a main conclusion of the paper is about responses to strain versus stress, shouldn't STA be performed with e_xx_, e_yy_, e_xy_ to differentiate those parameters? In this respect the question was raised why the authors focus on the annular pattern. Given the annular form of the partial slip and the high-correlation of deformations across the fingertip such a pattern is trivially expected (especially using ||e||). The authors are recommended to use the mentioned more specific parameters e_xx_, e_yy_, e_xy_, and focus first on the RF, and second on deformations elsewhere (maybe using some decorrelation techniques).

We thank the reviewers for this suggestion and we agree that it is valuable to look at the STA for more specific parameters. Instead of looking at *e_xx_*, *e_yy_* and *e_xy_* which are specific to a choice of reference frame (and are further analyzed with the linear regression model in Figure 6), we choose to use *e*_1_ and *e*_2_, the principal strains. The results are shown in a new Supplementary Figure 3, and are consistent with the linear model results. Indeed, while the STA of the compressive component matches the location of the RF, the STA of the tensile component is weaker and does not match the RF location.

We added a paragraph in the Results to cover this part. Given that the two approaches (STA and linear model) provide complementary but converging evidence, we did not estimate necessary to put this figure in the main text.

“The same STA analysis was repeated using the two principal strain components, one compressive and one tensile, separately to build two STA maps (see Materials and methods). […] Moreover, we found that the compressive STA peaks were generally larger and more often found in the afferent receptive field than their tensile counterpart, suggesting that FA-I afferents are more sensitive to compression. This finding will be further supported in the next section.”

Note, we do not mention at any point stresses, as we do not measure them. We only measure and discuss the strains. We suppose this comment is about stretch vs compression. Yes, indeed, an annular pattern is trivially expected IF the afferents encode something about the strains, as already mentioned in the manuscript. As the first goal of this STA analysis is to confirm the link between strain and spiking, it is reasonable to use it, and it explains the choice of the norm of the strain rates as well, which summarizes the strains in a single variable.

5. It was further suggested to attempt using GLMs or similar to capture non-linearities between strain and spikes. The rigid linear model will fail to capture expected non-linear relationships between spike rates and other parameters.

The modeling approach is not aimed at perfectly predicting the firing activity of the afferents, but rather uncover which aspects of the skin mechanics are encoded (i.e. using model selection). Building a more accurate firing model is a complex problem, given the afferent branching for instance but also the precise spike timing of the afferents; that would require a lot more data with more experimental conditions and a better temporal resolution, as already mentioned in the Discussion. We are actually currently trying this with new experiments.

6. Both reviewers found the part of the discussion stimulating, in which the authors touch upon orientation selectivity (line 351-354). It seems an interesting point to consider the differences between slips and edges – both in forces and neural information. Do the authors have evidence to think the brain could tell the difference between a slip and an edge? Alternatively, the speculation starts to explain orientation selectivity of primate tactile neurons in terms of prehension (rather than perception of edge orientation, as is typically assumed). This point is considered to be an outcome of their novel measurements. It should be worked out and presented more prominently.

An actual edge would be a more complete stimulus, given that it would also add an important local vertical compression (probably explaining SA-I strong sensitivity to edges), and its symmetrical nature (step up and then step down) would generate tangential compression and then stretch. Therefore, the comparison is only limited, and we do not want to speculate about it too much in the Discussion.